# *Mycobacterium tuberculosis* suppresses host antimicrobial peptides by dehydrogenating L-alanine

Cheng Peng [1,2,5], Yuanna Cheng[1,2,5], Mingtong Ma[1,2,5], Qiu Chen[1,2], Yongjia Duan[1,2], Shanshan Liu[1,2], Hongyu Cheng[1,2], Hua Yang[1,3], Jingping Huang[1,2], Wenyi Bu[1,2], Chenyue Shi[1,2], Xiangyang Wu[1,4], Jianxia Chen[1,3,4], Ruijuan Zheng[1,3], Zhonghua Liu[1,3], Zhe Ji[2], Jie Wang[1,3], Xiaochen Huang[1,3], Peng Wang[3], Wei Sha[3], Baoxue Ge [1,2,3,4] ✉ & Lin Wang [1,2,3] ✉

Antimicrobial peptides (AMPs), ancient scavengers of bacteria, are very poorly induced in macrophages infected by *Mycobacterium tuberculosis* (*M. tuberculosis*), but the underlying mechanism remains unknown. Here, we report that L-alanine interacts with PRSS1 and unfreezes the inhibitory effect of PRSS1 on the activation of NF-κB pathway to induce the expression of AMPs, but mycobacterial alanine dehydrogenase (Ald) Rv2780 hydrolyzes L-alanine and reduces the level of L-alanine in macrophages, thereby suppressing the expression of AMPs to facilitate survival of mycobacteria. Mechanistically, PRSS1 associates with TAK1 and disrupts the formation of TAK1/TAB1 complex to inhibit TAK1-mediated activation of NF-κB pathway, but interaction of L-alanine with PRSS1, disables PRSS1-mediated impairment on TAK1/TAB1 complex formation, thereby triggering the activation of NF-κB pathway to induce expression of AMPs. Moreover, deletion of antimicrobial peptide gene *β-defensin 4* (*Defb4*) impairs the virulence by Rv2780 during infection in mice. Both L-alanine and the Rv2780 inhibitor, GWP-042, exhibits excellent inhibitory activity against *M. tuberculosis* infection in vivo. Our findings identify a previously unrecognized mechanism that *M. tuberculosis* uses its own alanine dehydrogenase to suppress host immunity, and provide insights relevant to the development of effective immunomodulators that target *M. tuberculosis*.

Until the coronavirus (COVID-19) pandemic, tuberculosis (TB) was the leading cause of death from a single infectious agent, ranking above HIV/AIDS. In 2022, *M. tuberculosis* infection was responsible for estimated 10.6 million new TB cases and 1.3 million deaths[1]. *M. tuberculosis* infection usually triggers host innate and adaptive immune cells to restrict bacterial growth[2–5]. However, *M. tuberculosis* has responded to

these host defense strategies by evolving virulence factors to counteract host antibacterial mechanisms and facilitate successful intracellular infection[6–8].

Antimicrobial peptides (AMPs), small cationic and amphipathic peptides, are ancient members of the host defense system that act against a diverse set of pathogens[9,10]. It has been shown that the

[1]Shanghai Key Laboratory of Tuberculosis, Shanghai Pulmonary Hospital, Tongji University School of Medicine, Shanghai, China. [2]Department of Microbiology and Immunology, Tongji University School of Medicine, Shanghai, China. [3]Shanghai Clinic and Research Center of Tuberculosis, Shanghai Pulmonary Hospital, Tongji University School of Medicine, Shanghai, China. [4]Clinical Translation Research Center, Shanghai Pulmonary Hospital, Tongji University School of Medicine, Shanghai, China. [5]These authors contributed equally: Cheng Peng, Yuanna Cheng, Mingtong Ma. ✉e-mail: gebaoxue@sibs.ac.cn; 651377481@qq.com

expression of AMPs is induced by *M. tuberculosis*[11,12] and contributes to controlling its infection[13–16]. However, unlike the strong induction (20- to 40-fold) of AMPs expression by the extracellular pathogens, *Pseudomonas aeruginosa* and *Streptococcus*, in macrophages[17], infection by the intracellular pathogens, *M. tuberculosis* or *Mycobacterium avium*, induces very low levels of antimicrobial peptide *Hepcidin* mRNA in bone marrow-derived monocytes (BMDMs) or the human monocytic cell line THP1[18]. Similarly, in vitro experiments of *M. tuberculosis* infection have shown that β-defensin is only induced at a high multiple of infection (MOI) in alveolar macrophages and is not detected in blood monocytes at any experimental MOI[19,20]. Consistent with this, it has been shown that AMPs are not detected in tuberculous granulomas[19–21]. This suggests that there may be an additional mechanism underlying the suppression of AMPs by pathogenic mycobacteria in monocytes or macrophages, which serve as both habitats for, and the first line of defense against *M. tuberculosis*.

One striking characteristic of *M. tuberculosis* is its utilization of type VII secretion systems to secrete numerous proteins across its hydrophobic and highly impermeable cell walls[22]. However, it has remained unclear whether and how such *M. tuberculosis*-secreted proteins inhibit the production of AMPs. It has been shown that *M. tuberculosis* infection can induce AMPs efficiently in a human lung epithelial A549 cell line and respiratory murine epithelial cells[19,20]. Considering that epithelial cells have very limited phagocytic capacity compared with macrophages, we hypothesized that *M. tuberculosis* inhibited the production of AMPs in macrophages through their secretory proteins.

In this work, by screening of *M. tuberculosis* secretory proteins that inhibit the expression of antimicrobial peptide *DEFB4* in HEK293T cells and knockout strain validation in macrophages, we observe that *M. tuberculosis* alanine hydrogenase Rv2780 inhibits the expression of AMPs. Mechanistically, we find that Rv2780 dehydrogenates L-alanine and reduces the level of L-alanine in macrophages. By streptavidin-biotin-L-alanine pull down assay, we show that L-alanine interacts with PRSS1. Moreover, L-alanine relieves the inhibitory effects of PRSS1 on NF-κB activation to induce the expression of AMPs. Functionally, both supplementation of L-alanine and Rv2780 inhibitor GWP-042 show inhibitory activity against *M. tuberculosis* infection in macrophages and in vivo.

## Results

### Rv2780 inhibits the expression of AMPs

HEK293T cells are widely used to study the function of pathogenic bacteria secretory proteins on the activation of host NF-κB and MAPKs signal, indicating the existence of integral immune molecules of these two pathways in HEK293T cells[23,24]. We also detected endogenous *β-Defensin 4* (*DEFB4*) mRNA levels in HEK293T cells to verify AMP *DEFB4* expression at baseline (Supplementary Fig. 1A). To identify *M. tuberculosis* proteins that inhibit the expression of AMPs, we transfected HEK293T cells with plasmids encoding 201 *M. tuberculosis* secreted proteins or lipoproteins[25] and examined their effects on the expression of *DEFB4* using reverse transcription (RT)-PCR (Supplementary Fig. 1B; Supplementary Data 1). Rv2780, a secreted alanine dehydrogenase[26,27] of *M. tuberculosis*, was found to reduce the mRNA levels of several AMPs including not only *DEFB4* but also *β-Defensin 3* (*DEFB3*) and *Cathelicidin Antimicrobial Peptide* (*CAMP*), as measured by RT-PCR assay (Supplementary Fig. 1C–E). Rv2780 was detected in both the supernatants and lysates of *M. tuberculosis* cultures (Supplementary Fig. 1F, G), illustrating that Rv2780 is a secreted protein. In addition, Rv2780 was detected in the cytoplasm of mice peritoneal macrophages (MPMs) and A549 cells during *M. tuberculosis* infection (Supplementary Fig. 1H, I), suggesting Rv2780 could be secreted to host cells. However, compared to A549 cells, much more abundant Rv2780 protein was detected in H37Rv-infected macrophages (Supplementary Fig. 1H), suggesting a more powerful function of Rv2780 in

macrophages. To analyze the subcellular localization of Rv2780 during *M. tuberculosis* infection, we detected Rv2780 by immunofluorescence microscopy. Rv2780 was mainly detected in the cytoplasm, partially in mitochondria, very minimally in the endoplasmic reticulum (ER) or lysosome (Supplementary Fig. 1I).

To further evaluate whether Rv2780 inhibits the expression of AMPs during *M. tuberculosis* infection, we deleted Rv2780 from an *M. tuberculosis* H37Rv strain, thus generating an H37RvΔRv2780 strain (Supplementary Fig. 1F, G). Consistent with previous report[28,29], Rv2780 did not significantly change in vitro H37Rv growth in aerobic condition or fitness to hypoxic condition (Supplementary Fig. 1J, K). Electronic scanning microscopy analysis showed the similar morphology of H37RvΔRv2780 and H37Rv strain (Supplementary Fig. 1L). Rv2779c is an Lrp/AsnC family transcriptional factor that binds amino acid ligands to regulate *Rv2780* expression[30,31]. Deletion of Rv2780 in H37Rv strain dramatically decreased *Rv2780* expression but did not significantly change *Rv2779c* expression (Supplementary Fig. 1M, N). Besides, alanine level was significantly increased in H37RvΔRv2780 strain (Supplementary Fig. 1O), suggesting that Rv2780 may function as an alanine dehydrogenase in *M. tuberculosis*.

Macrophages, which serve as both habitats for and the first line of defense against *M. tuberculosis*, were infected with the H37Rv or H37RvΔRv2780 strain. Primary peritoneal macrophages infected with H37Rv showed limited increase in the expression of *Defb4* (9.63-fold), *Defb3* (5.67-fold) and *Camp* (3.79-fold) at 24 h post-infection (Fig. 1A and Supplementary Fig. 2A, B). However, H37RvΔRv2780 was associated with much higher induction of the mRNA of *Defb4* (21.08-fold), *Defb3* (16.94-fold) and *Camp* (10.41-fold) than in cells infected with wild-type H37Rv for 24 h (Fig. 1A and Supplementary Fig. 2A, B). Complementation of H37RvΔRv2780 with Rv2780 restored the ability of *M. tuberculosis* to suppress the expression of *Defb4*, *Defb3* and *Camp* (Fig. 1B and Supplementary Fig. 2C, D). Taken together, these results suggest that *M. tuberculosis* Rv2780 may inhibit the expression of AMPs.

Antimicrobial peptides kill bacteria directly in vitro and are crucial for macrophages to limit the intracellular survival of *M. tuberculosis*[11–15]. We also examined direct killing effects of AMPs on *M. tuberculosis* as described previously by ref. 11, and found that the MIC of Defb4, Defb3 and Camp were at 0.01 μg/ml, 10 μg/ml and 0.1 μg/ml, respectively, suggesting that these AMPs may have the anti-*M. tuberculosis* activity in vitro (Supplementary Fig. 2E). To examine whether Rv2780 regulates the intracellular survival of *M. tuberculosis*, we infected primary peritoneal macrophages with H37Rv or H37RvΔRv2780 strains and measured the survival rate of intracellular *M. tuberculosis* using a colony forming unit (CFU) assay. H37RvΔRv2780 showed much lower CFU counts in macrophages at 24-h post-infection than H37Rv and H37Rv(ΔRv2780 + Rv2780) (Fig. 1C, D), suggesting that Rv2780 may be essential for the intracellular survival of *M. tuberculosis*. ROS production and xenophagy were also shown to restrict the intracellular *M. tuberculosis*[32], however deletion of Rv2780 did not significantly change ROS production and xenophagy during *M. tuberculosis* infection in macrophages (Supplementary Fig. 2F–H). H37RvΔRv2780 infected macrophages had much lower levels of mRNAs encoding proinflammatory cytokines Interleukin (IL)−1β, IL-6, IL-12p40 and Tumor Necrosis Factor α (TNFα) (Supplementary Fig. 2I–L).

To further investigate the functional relevance of Rv2780 in the in vivo pathogenesis of *M. tuberculosis* infection, we challenged C57BL/6J mice with wild-type H37Rv, H37RvΔRv2780 or H37Rv(ΔRv2780 + Rv2780) for 30 days. The bacterial burden in the lung tissues of mice infected with H37RvΔRv2780 was much lower (decreased 1.26-fold in $\log_{10}$) than mice infected with H37Rv and H37Rv(ΔRv2780 + Rv2780) (Fig. 1E). Consistent with this, lung tissues from mice infected with H37RvΔRv2780 showed less immune-cell infiltration and fewer inflammatory lesions than those from mice infected with H37Rv (Fig. 1F, G). The lung tissue of mice infected with H37RvΔRv2780 exhibited much lower expression of *Il1b*, *Il6*, *Il12* and *Tnf* than the lung

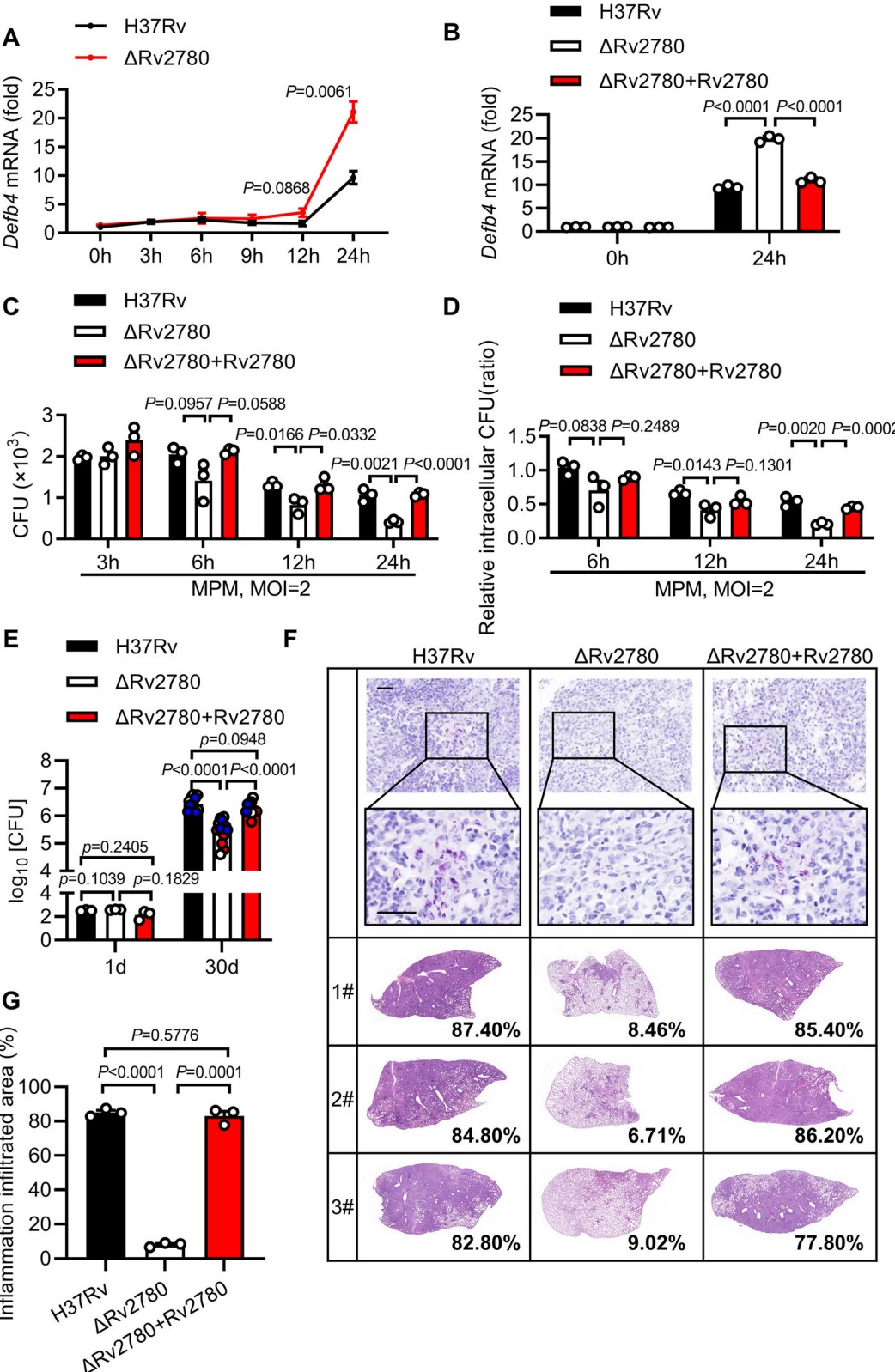

tissue of mice infected with H37Rv (Supplementary Fig. 2M–P). Together, these results suggest that Rv2780 is an essential virulence factor of *M. tuberculosis*.

## Rv2780 dehydrogenates L-alanine

Rv2780 encodes L-alanine dehydrogenase, an enzyme that catalyzes the NAD⁺-dependent interconversion of alanine and pyruvate[26,27]

(Fig. 2A). The enzymatic kinetics of Rv2780 was assesses by analyzing the enzymatic product pyruvate. The $K_m$ and $V_{max}$ were found to be 0.964 mM and 111.8 M/s, respectively (Fig. 2B). Another in vitro alanine dehydrogenation assay showed that the addition of purified recombinant wild-type Rv2780 led to the greater production of NADH from alanine (Fig. 2C), suggesting that Rv2780 has the alanine dehydrogenase activity.

**Fig. 1 | Rv2780 is a virulence factor. A** RT-PCR analysis of *Defb4* in mice peritoneal macrophages infected with wild-type H37Rv or H37RvΔRv2780 for 0, 3, 6, 9,12 and 24 h (multiplicity of infection (MOI) = 2). **B** RT-PCR analysis of *Defb4* in mice peritoneal macrophages infected with wild-type H37Rv, H37RvΔRv2780 or H37RvΔRv2780 + Rv2780 for 0 and 24 h (MOI = 2). Intracellular colony-forming units (CFUs) assay. CFU counts (**C**) and relative intracellular CFU ratio (**D**) in mice peritoneal macrophages (MPMs) infected with wild-type H37Rv, H37RvΔRv2780 and H37RvΔRv2780 + Rv2780 for 3, 6, 12, and 24 h (MOI = 2). 6–8 weeks old female C57BL/6J mice were aerosol-infected with roughly 200 CFUs per mouse of H37Rv, H37RvΔRv2780 or H37RvΔRv2780 + Rv2780. We assayed: CFU of bacterial load (**E**); lung sections with acid-fast staining (**F**), haematoxylin and eosin staining (**F**) and histological score (**G**). Data in (**A–E** and **G**) are representative of one experiment with at least three independent biological replicates; (**A–D**) $n = 3$, each circle represents one technical repeat (mean ± s.e.m); (**E**) $n = 3$ mice infected for 1 day and $n = 12$ mice infected for 30 days with red, blue and white circles denoting separate experiments (mean ± s.e.m); (**G**) $n = 3$ mice (mean ± s.e.m). Two-tailed unpaired Student's *t*-test (**A–D**) and two-sided Mann-Whitney *U*-test (**E**, **G**) were used for statistical analysis. *P* values are shown in (**A–D** and **E**, **G**). 1#, 2# and 3# in (**F**) represent lung tissues from 3 mice infected for 30 days. Scale bars, 100 μm (top; original magnification, ×400) and 20 μm (bottom; original magnification, ×1000). Source data are provided as a Source Data file.

By performing gas chromatography-mass spectroscopy analysis of metabolites in sera of C57BL/6J mice infected with *M. tuberculosis* H37Rv, we found that the level of alanine was markedly reduced in sera of infected mice (Fig. 2D; Supplementary Data 2). By contrast, other amino acids such as methionine, phenylalanine and aspartic acid were not significantly changed in response to H37Rv infection (Fig. 2E and Supplementary Fig. 3A–C), suggesting that the decreased alanine level may be specifically caused by *M. tuberculosis* infection rather than food intake or metabolism. Moreover, smear-positive patients with TB had much lower level of alanine in their plasma than healthy people (Fig. 2F). This is consistent with a previous report showing that alanine was one of the metabolites showing the greatest decrease in a [1]H nuclear magnetic resonance spectroscopy-based metabolomic analysis of sera from TB patients[33]. Host glutamic pyruvic transaminase (GPT) also known as alanine aminotransferase (ALT) can catalyze the reversible interconversion of L-alanine and 2-oxoglutarate to pyruvate and L-glutamate[34]. Therefore, we next analyzed the relationship between alanine level and GPT in sera of TB patients. However, as shown in Supplementary Fig. 3D–F, no significant correlation between alanine and GPT was noted in patients with TB. Together, the decrease of alanine level in *M. tuberculosis*-infected mice and TB patients might be mediated by *M. tuberculosis* infection.

We further compared the chest X-ray score and the smear score between the top seven patients with the highest plasma alanine level and the bottom seven patients. We found TB patients with lower alanine level exhibited a trend of more severe pulmonary pathological damage, indicated by higher X-ray score (Supplementary Fig. 3F, G). However, it seems that alanine level is not correlated with the smear score (Supplementary Fig. 3H). This may be because the smear score cannot fully reflect the bacterial load in TB patients.

Structural analysis of Rv2780 revealed two typical alanine dehydrogenase activity sites at histidine 96 (H[96]) of the catalytic domain and aspartic acid 270 (D[270]) of the NAD⁺ binding domain, which are highly conserved across different bacterial species (Supplementary Fig. 3I). Mutation of two active sites on Rv2780 (Rv2780[DM], with H96A and D270A) impaired its alanine dehydrogenase activity (Fig. 2C). Overexpression of wild-type Rv2780, but not its inactive mutant Rv2780[DM] markedly decreased the level of L-alanine in both HEK293T and A549 cells (Supplementary Fig. 3J, K). Moreover, the level of alanine was reduced in H37Rv or H37Rv(ΔRv2780 + Rv2780) infected macrophages, but infection of H37RvΔRv2780 or H37Rv(ΔRv2780 + Rv2780[DM]) led to much more abundant alanine in the infected cells (Fig. 2G, H; Supplementary Fig. 3L). We analyzed total metabolic profiling of macrophages infected with H37Rv, H37Rv(ΔRv2780), H37Rv(ΔRv2780 + Rv2780) and H37Rv(ΔRv2780 + Rv2780[DM]) for 24 h. Consistently, the level of pyruvate was higher in H37Rv or H37Rv (ΔRv2780 + Rv2780) infected macrophages than that of H37RvΔRv2780 or H37Rv (ΔRv2780 + Rv2780[DM]) infection group (Supplementary Fig. 3L; Supplementary Data 3). However, deletion Rv2780 had no effects on the content of other metabolites in central carbon metabolism, including citrate, α-ketoglutarate, fumarate, and lactate (Supplementary Fig. 3M, N; Supplementary Data 3).

To measure the effect of Rv2780 on central carbon metabolic flux, [13]C-labeled tracing analysis was conducted in macrophages. As shown in Supplementary Fig. 4A–D, [13]C6-glucose is converted to produce labeled pyruvate, which can be decarboxylated to form labeled two-carbon metabolite acetyl-CoA and entered into Trichloroacetic acid (TCA) cycle, while unlabeled alanine is dehydrogenized by Rv2780 to form unlabeled pyruvate. Pyruvate provides two carbons to acetyl-CoA, citrate, α-ketoglutarate, succinate and fumarate. Both labeled and unlabeled pyruvate-derived acetyl-CoA are entered into the TCA cycle respectively. Compared with ΔRv2780 infection, the percentage of unlabeled alanine was decreased in H37Rv-infected macrophage suggesting the metabolic flux from alanine to pyruvate in the presence of Rv2780 (Supplementary Fig. 4B; Supplementary Data 4). However, the percentage of unlabeled other metabolites were not significantly different between H37Rv- and H37RvΔRv2780-infected macrophages. These results suggest that Rv2780 has no significant effect on macrophages glycolysis or TCA cycle.

Besides, complementation of H37RvΔRv2780 with wild-type Rv2780, rather than Rv2780[DM] mutant significantly decreased alanine in lung tissues and sera from H37Rv infected mice at 7- and 28-days post-infection (Fig. 2I, J). These results suggest that *M. tuberculosis* may have evolved a metabolic ability to dehydrogenate L-alanine via Rv2780 in host cells and can therefore reduce the alanine level in eukaryotes.

### Rv2780 suppresses AMPs by dehydrogenating alanine

Given that Rv2780 decreased both the level of L-alanine and expression of AMPs, we hypothesized Rv2780 might suppress AMPs expression through L-alanine dehydrogenation. To verify the effect of L-alanine on AMPs expression, we supplemented macrophages with L-alanine before *M. tuberculosis* infection. Addition of L-alanine significantly increased mRNA levels of *Defb4* (27.95-fold), *Defb3* (9.97-fold) and *Camp* (8.57-fold) in macrophages infected with *M. tuberculosis* H37Rv for 24 h (Fig. 3A and Supplementary Fig. 5A, B). Rv2780 also shows glycine dehydrogenase activity in vitro[35,36]. We supplemented Rv2780-overexpressed HEK293T cell with L-alanine or glycine, and ELISA analysis was performed to determine the protein level of Defb4 and Camp[37,38]. Administration of alanine rather than glycine rescued the Rv2780-mediated inhibition of AMPs expression (Supplementary Fig. 5C–E). Moreover, only supplementation with L-alanine, but not D-alanine or glycine increased Defb4 and Camp protein level in response to H37Rv infection (Fig. 3B and Supplementary Fig. 5F). These results suggest that Rv2780 may inhibit AMPs expression through its alanine dehydrogenase activity.

To further examine whether Rv2780 suppresses the AMPs by its dehydrogenase activity, we infected mice peritoneal macrophages or BMDMs with H37Rv(ΔRv2780 + Rv2780) or H37Rv(ΔRv2780 + Rv2780[DM]) and examined the protein level or mRNA level of *Camp* and *Defb4*. Only the H37RvΔRv2780 strain complemented with wild-type Rv2780, but not with Rv2780[DM], restored the ability of *M. tuberculosis* to suppress Defb4 and Camp (Fig. 3C and Supplementary Fig. 5G-J). Consistently, infection with H37Rv(ΔRv2780 + Rv2780), but not with H37Rv(ΔRv2780 + Rv2780[DM]), induced much lower production of Camp and Defb4 in the serum or lung of mice (Fig. 3D, E and Supplementary Fig. 5K, L).

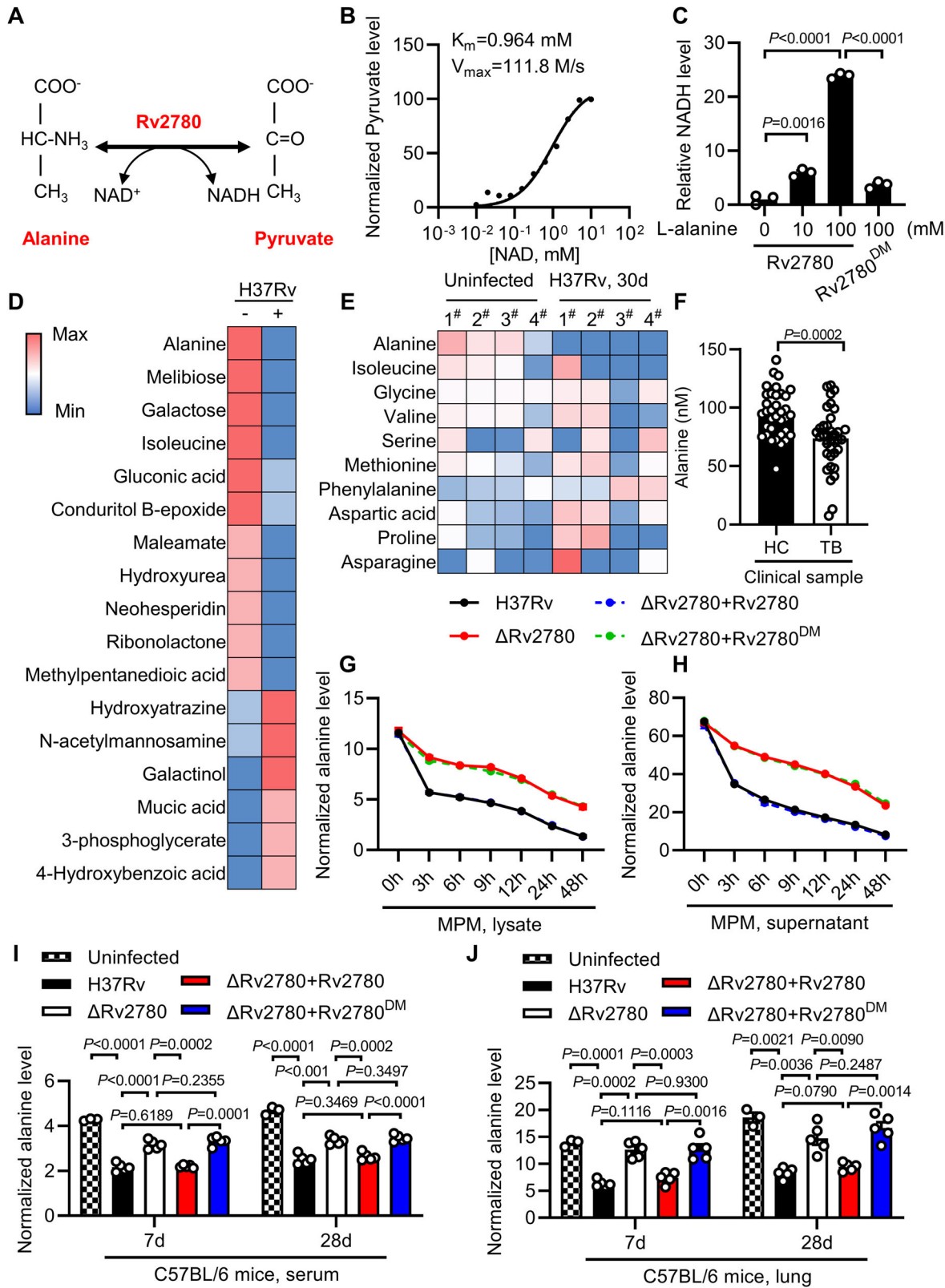

H37Rv(ΔRv2780 + Rv2780), but not H37Rv(ΔRv2780 + Rv2780^DM), rescued an Rv2780-mediated increase in the intracellular survival of *M. tuberculosis* in mice peritoneal macrophages (Fig. 3F, G). However, no difference in cell viability was observed in macrophages infected with different strains (Supplementary Fig. 6A). In addition, deletion of Rv2780 also reduced *M. tuberculosis* survival in BMDMs (Supplementary Fig. 6B, C), alveolar macrophages (Supplementary Fig. 6D, E), or

neutrophils (Supplementary Fig. 6F, G), and the reduced survival of H37RvΔRv2780 was rescued by the complementation of Rv2780, but not Rv2780^DM. These results suggest that *M. tuberculosis* Rv2780 may suppress the expression of AMPs, thus promoting *M. tuberculosis* intracellular survival by its alanine dehydrogenase activity.

To examine whether Rv2780 increased *M. tuberculosis* intracellular survival via inhibiting AMPs, we infected *Defb4^-/-* mice peritoneal

**Fig. 2 | Rv2780 dehydrogenates L-alanine. A** Catalytic model diagram: Rv2780 is an alanine dehydrogenase. Nicotinamide adenine dinucleotide (NAD), reduced NAD (NADH). **B** Dose-response curves of Rv2780 detected by pyruvate. Km, Vmax were shown in the figure. **C** In vitro catalytic assay of Rv2780 with increasing concentration of L-alanine (0, 10, 100 mM). Rv2780 enzyme inactive mutant Rv2780 (H96A, D70A) (Rv2780DM) was used as negative control. **D** Heat map of downregulated (blue) and upregulated (red) metabolites in sera from C57BL/6J mice aerosol-infected with roughly 200 CFUs per mouse of H37Rv for 30 days; Uninfected $n = 4$, H37Rv infection $n = 4$. **E** Heat map of all the detected amino acids in sera from each uninfected or H37Rv infected mice. 1#, 2#, 3# and 4# represent serum samples from 4 mice. **F** Quantitative analysis of alanine in plasma from healthy controls (HC) or tuberculosis patients (TB) (mean ± s.e.m. of $n = 35$). Alanine detection assay. Normalized alanine level in cell lysates (**G**) and supernatants (**H**) of MPMs infected with wild-type H37Rv, H37RvΔRv2780, H37RvΔRv2780 + Rv2780 or H37RvΔRv2780 + Rv2780 DM for 0, 3, 6, 9,12, 24 and 48 h (MOI = 2). Normalized alanine level in sera (**I**) or lung homogenates (**J**) of mice infected with indicated strains for 7 and 28 days (mean ± s.e.m. of $n = 3$ or $n = 5$). Alanine level was normalized to GAPDH level in (**G** and **H**). Alanine level was normalized to protein level in (**I**) and (**J**). Data in (**B**, **C**) and (**F**, **G**–**J**) are representative of one experiment with at least three independent biological replicates; **C** $n = 3$, each circle represents one technical repeat (mean ± s.e.m.); **F** $n = 35$ samples for each group (mean ± s.e.m.); (**I**, **J**) $n = 4$ mice in uninfected group or $n = 5$ mice in other groups (mean ± s.e.m.). Two-tailed unpaired Student's $t$-test (**C**) and two-sided Mann-Whitney $U$-test (**F**, **I**, **J**) were used for statistical analysis. $P$ values are shown in (**C**, **F**, **I** and **J**). Source data are provided as a Source Data file.

macrophage with H37Rv, H37RvΔRv2780, H37Rv(ΔRv2780 + Rv2780) and H37Rv(ΔRv2780 + Rv2780DM). We found that deletion of Defb4 markedly increased intracellular survival of H37Rv, and eliminated the enhanced effects of Rv2780 on intracellular survival of H37Rv and H37Rv(ΔRv2780 + Rv2780) (Fig. 3H, I). Moreover, we infected Defb4 knockout mice with H37Rv or H37RvΔRv2780 to further validate in vivo relevance of Rv2780 and Defb4. Knockout of Defb4 markedly increased bacterial burden and pathological damages in lung tissues of the *M. tuberculosis* H37Rv-infected mice, and abolished the increased bacterial burden and pathological damages by Rv2780 in lung tissues of the *M. tuberculosis*-infected mice (Fig. 3J–L). However, no difference in the alanine level in sera and lung tissues was detected between WT and *Defb4*⁻/⁻ mice when infected with H37Rv (Supplementary Fig. 6H, I). Above all, these data suggest that Rv2780 may increase the survival of *M. tuberculosis* through suppressing the expression of AMPs.

## L-Alanine interacts with PRSS1

We next investigated the mechanism underlying the induction of AMPs by L-alanine. By performing biotin-streptavidin pull-down assay combined with mass spectrometry analyses[39,40] (Fig. 4A and Supplementary Fig. 7A; Supplementary Data 5), we found that cationic trypsinogen (protease serine 1, PRSS1), encoded by a susceptibility gene associated with chronic pancreatitis[41], interacted with L-alanine, but not with D-alanine (Fig. 4B, C), and non-biotinylated L-alanine could competitively elute biotinylated L-alanine from PRSS1 (Supplementary Fig. 7B). PRSS1 is a serine protease composed of the N-terminal alpha-trypsin chain 1 and C-terminal chain 2 that are linked by a disulfide bond[42,43] (Supplementary Fig. 7C). Only the N-terminal alpha-trypsin chain 1, not C-terminal chain 2, of PRSS1 interacted with L-alanine (Supplementary Fig. 7D). MicroScale Thermophoresis (MST) analysis[44] revealed that L-alanine strongly interacted with PRSS1 (KD = $8.88 \times 10^{-5}$ M) (Fig. 4D). These results suggest that L-alanine may interact with PRSS1.

It has been shown that *M. tuberculosis* infection induces the expression of AMPs through the TLR2/NF-κB signaling pathway[11]. Upon stimulation, the ubiquitin ligase, TRAF6, which is downstream of the TLR2 receptor, induces TAK1 oligomerization-dependent auto-phosphorylation and TAK1 subsequently activates the IKK-mediated NF-κB signaling pathway[45,46]. We next examined whether PRSS1 had any effect on activation of NF-κB using a luciferase reporter gene assay. As shown in Supplementary Fig. 7E, the overexpression of PRSS1 markedly suppressed the activation of NF-κB by TRAF6 or TAK1, but not that mediated by IKKα/β, suggesting that PRSS1 may block the activation of NF-κB signaling by acting at downstream of the TAK1 complex and upstream of IKKα/β.

To elucidate the mechanism underlying the inhibition of NF-κB signaling by PRSS1, we examined the interactions between PRSS1 and TLR pathway signaling molecules. PRSS1 was found to interact with TAK1, which is co-expressed with TAB1 in HEK293T cells (Fig. 4E). The interaction between TAK1 and TAB1 is important for the activation of

TAK1[47]. In HEK293T cells, PRSS1 markedly impeded the interaction between TAK1 and TAB1 and consequently inhibited the enhanced phosphorylation of TAK1 by TAB1 (Fig. 4F, G). Moreover, enhanced formation of TAK1-TAB1 complex was found in *Prss1*⁺/⁻ peritoneal macrophages (Fig. 4H), suggesting that PRSS1 may disrupt formation of the TAK1-TAB1 complex. Lastly, deletion of Rv2780 markedly increased the phosphorylation of p65, but treatment of TAK1 inhibitor ((5Z)-7-oxozeaenol, 5Z-7Ox) eliminated the reduced phosphorylation of p65 by Rv2780. (Supplementary Fig. 7F). Consistently, inhibition of p65 phosphorylation by Rv2780 was not observed in *Prss1*⁺/⁻ macrophages (Fig. 4I). Much higher level of NF-κB activation and AMPs expression are also observed in *Prss1*⁺/⁻ macrophages in response to gram-negative bacteria *Escherichia coli* (*E. coli*) (Supplementary Fig. 7G) or another gram-positive bacteria *Staphylococcus aureus* (*S. aureus*) infection (Supplementary Fig. 7H), suggesting the inhibition of NF-κB by PRSS1 may be a general mechanism. These results suggest that Rv2780 may inhibit NF-κB signaling via PRSS1 and TAK1 during *M. tuberculosis* infection.

## L-alanine induces AMPs via PRSS1

We next investigated the role of PRSS1 in the regulation of AMPs. *Prss1*⁺/⁻ macrophages had much higher mRNA levels of AMPs than wild-type cells infected with *E. coli* (Supplementary Fig. 7I–K), *S. aureus* (Supplementary Fig. 7I–N), or *M. tuberculosis* H37Rv (Fig. 4J and Supplementary Fig. 7O–Q), suggesting PRSS1 is a potent negative regulator of AMPs expression.

The in vivo role of *Prss1* in macrophages was validated by generating macrophage conditional *Prss1* knockout mice (*Lyz2*cre*Prss1*floxp/floxp mice). Accordingly, *Lyz2*cre*Prss1*floxp/floxp mice exhibited decreased lung bacterial burden and tissue damage compared with *Prss1*floxp/floxp mice (Fig. 4K–M). These results suggest that PRSS1 may inhibit the induction of AMPs, and negatively regulates anti-TB immunity.

To further examine the functional relevance of PRSS1 and L-alanine, peritoneal macrophages from WT or *Prss1*⁺/⁻ mice were treated with L-alanine followed by infection with *M. tuberculosis* H37Rv. As shown in Fig. 4J and Supplementary Fig. 7O-Q, L-alanine promoted AMPs expression in WT but not *Prss1*⁺/⁻ MPMs, suggesting L-alanine induces AMPs via PRSS1. The binding affinity between PRSS1 and L-alanine was $8.88 \times 10^{-5}$ M, suggesting a strong interaction. To determine the threshold on L-alanine level to enhance antimicrobial peptide through PRSS1, we supplemented WT and *Prss1*⁺/⁻ macrophages with different concentrations of L-alanine, and found 0.01 mM L-alanine was sufficient to induce *Defb4* expression in WT, but not *Prss1*⁺/⁻ MPMs (Supplementary Fig. 7Q).

In addition, *Prss1*⁺/⁻ MPMs infected with *M. tuberculosis* H37Rv had much lower intracellular CFU than those WT counterparts (Fig. 4N, O). L-alanine significantly inhibited the intracellular survival of *M. tuberculosis* H37Rv in WT macrophages, but not in *Prss1*⁺/⁻ peritoneal macrophages (Fig. 4N, O), suggesting that L-alanine may restrict the intracellular growth of *M. tuberculosis* through PRSS1.

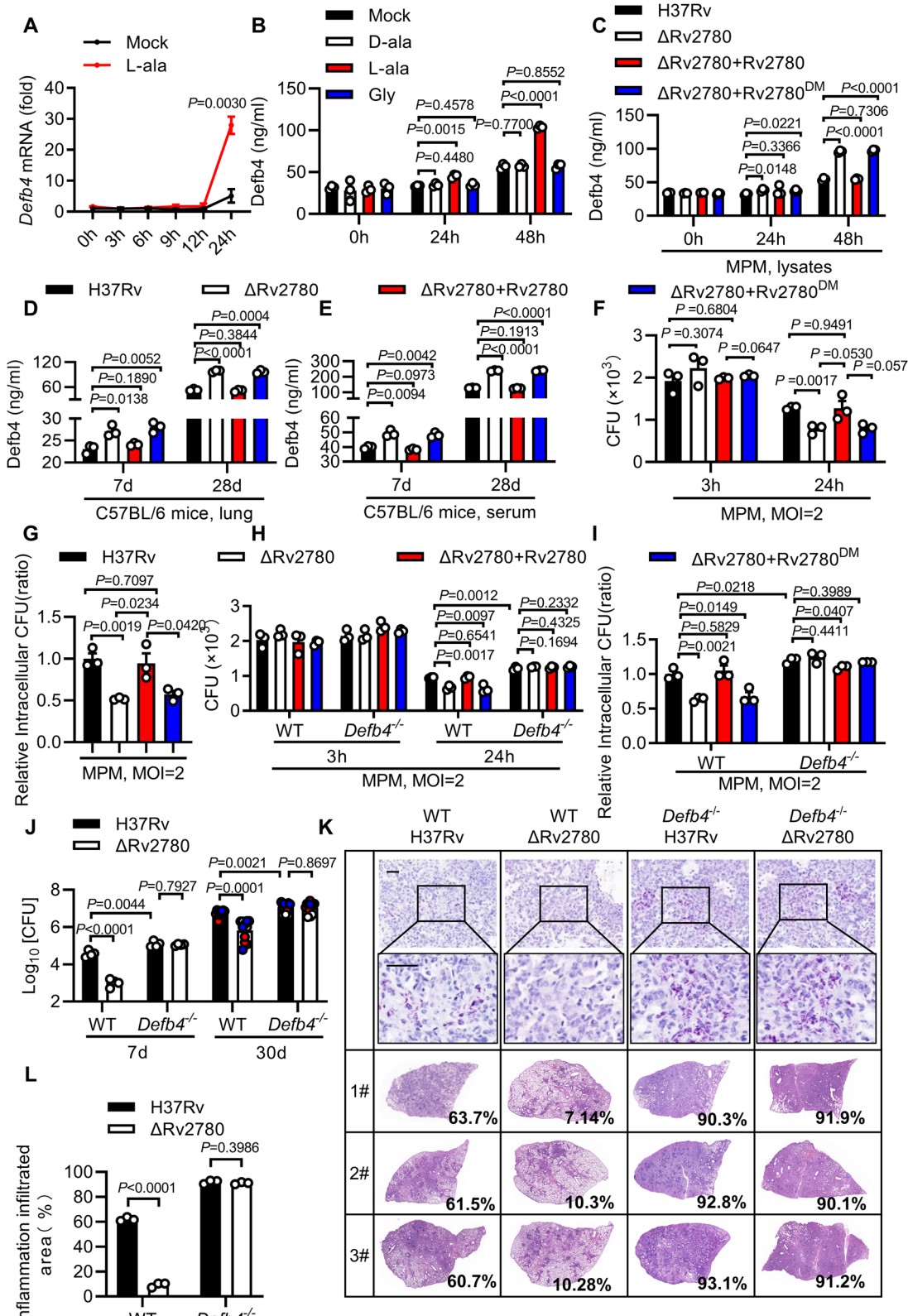

## Supplementation of L-alanine enhances anti-TB immunity

Above all, we aim to test the effect of L-alanine on the clearance of *M. tuberculosis* inside macrophages. The growth of the *M. tuberculosis* H37Rv strain in vitro was not significantly affected by L-alanine treatment (Supplementary Fig. 8A, B). However, treatment with L-alanine dramatically inhibited the intracellular survival of *M. tuberculosis* H37Rv at an efficient level equivalent to that of the best-in-class

antibiotic rifampicin (RIF)[48] and a combination of L-alanine and RIF resulted in an even lower bacterial burden compared with either agent alone (Fig. 5A, B), suggesting that L-alanine could be used to complement first-line anti-TB drugs. Moreover, L-alanine efficiently killed a clinical multiple-drug-resistant (MDR) *M. tuberculosis* strain in macrophages (Fig. 5C, D). No significant effect on cell viability was observed of L-alanine, no matter with or without H37Rv infection

**Fig. 3 | Rv2780 suppresses AMPs by dehydrogenating alanine. A** RT-PCR analysis of *Defb4* in MPMs treated with 1mM L-alanine (L-ala) for 12 h followed by H37Rv infection for another 0, 3, 6, 9, 12, and 24 h (MOI = 2). **B** ELISA analysis of Defb4 in MPMs treated with 1mM L-alanine, 1 mM D-alanine (D-ala) or 1 mM Glycine (Gly) for 12 h followed by H37Rv infection for another 0, 24 and 48 h (MOI = 2). ELISA analysis of Defb4 in cell lysates of MPMs (**C**) infected with indicated strains for 0, 24 and 48 h (MOI = 2) or lung homogenates (**D**) and sera (**E**) of mice infected with indicated strains for 7 and 28 days. CFU counts (**F**) and relative intracellular CFU ratio (**G**) in MPMs infected with indicated strains for 3 and 24 h (MOI = 2). CFU counts (**H**) and relative intracellular CFU ratio (**I**) in wild type and *Defb4^/-^* MPMs infected with indicated strains for 3 and 24 h (MOI = 2). CFU assay (**J**), histopathological assay by acid-fast staining and haematoxylin and eosin staining (**K**) and histological score (**L**)

in lung tissues of wild type (WT) and *Defb4^-/-^* mice aerosol-infected with H37Rv and H37RvΔRv2780 for 7 days and 30 days. Data in (**A–J**) and (**L**) are representative of one experiment with at least three independent biological replicates; (**A–C** and **F–I**) *n* = 3, each circle represents one technical repeat (mean ± s.e.m); (**D, E**) *n* = 3 mice (mean ± s.e.m). (**J**) *n* = 3 mice infected for 7 days and *n* = 12 mice infected for 30 days with red, blue and white circles denoting separate experiments (mean ± s.e.m). (**L**) *n* = 3 mice (mean ± s.e.m). Two-tailed unpaired Student's *t*-test (**A–C, F–I**) and two-sided Mann-Whitney *U*-test (**D, E, J, L**) were used for statistical analysis. *P* values are shown in **A–J** and **L**. 1#, 2# and 3# in (**K**) represent lung tissues from 3 mice infected for 30 days. Scale bars, 100 μm (top; original magnification, ×400) and 20 μm (bottom; original magnification, ×1000). Source data are provided as a Source Data file.

(Supplementary Fig. 8C, D). These results suggest that L-alanine may act as an efficient host-directed inhibitor of *M. tuberculosis*, particularly for drug-resistant *M. tuberculosis* for which current antibiotics are largely ineffective.

Since L-alanine was a strong inducer of AMPs that restrict the intracellular survival of *M. tuberculosis*, while *M. tuberculosis* infection substantially reduced the level of alanine in host immune cells, we next addressed the therapeutic effectiveness of L-alanine in vivo. In severe combined immunodeficient (SCID) mice model[49], mice given L-alanine lived much longer, suggesting L-alanine functions in an innate immunity-dependent way (Fig. 5E). C57BL/6J mice challenged with H37Rv were given double-distilled water or that containing 30 mg/mL L-alanine or D-alanine, and their lungs examined by histopathology and for bacterial burden. Upon *M. tuberculosis* H37Rv infection, mice supplemented with L-alanine, but not D-alanine, had less histological damage in their lungs than mice given double-distilled water alone (mock) (Fig. 5F–H). Similarly, the bacterial burden in the lungs of H37Rv-infected mice treated with L-alanine was also much lower (decreased 1.332-fold in log$_{10}$) than control mice. These results suggest that L-alanine may inhibit the pathogenesis of *M. tuberculosis* infection in vivo.

**Targeting Rv2780 inhibits the growth of mycobacteria in vivo**

The crystal structure of the *M. tuberculosis* Rv2780 (PDB code: 2VHX) with NAD$^+$ binding domain was used for structure-based virtual screening of commercial databases (Locator Library and MCE Compound Library), which contain 309,800 inhibitors. As shown in Fig. 6A–C, a small-molecule compound, (S)-N-(5-(3-fluorobenzyl)-1H-1,2,4-triazol-3-yl) tetrahydrofuran-2-carboxamide (GWP-042), bound to Rv2780, forming four hydrogen bonds, one cation - π interaction and multiple hydrophobic interactions (Supplementary Data 6). Localized surface plasmon resonance (SPR) assay revealed that GWP-042 interacted strongly with Rv2780. The equilibrium dissociation constant (KD) of GWP-042 to Rv2780 was 1.896×10$^{-5}$ M, nearly 3-10 folds lower than other reported anti-tuberculosis drug to their target protein[50,51] (Fig. 6D). To further clarify whether GWP-042 inhibits the activity of Rv2780, we measured the hydrogenase activity of Rv2780 in the presence of increasing concentrations of GWP-042. By measuring the enzymatic production of pyruvate that reflects the enzyme activity of Rv2780, the IC$_{50}$ of GWP-042 on Rv2780 was 0.21 ± 0.05 μM as indicated by pyruvate (Fig. 6E), which is almost 100 folds lower than the reported Rv2780 inhibitors[52]. These data suggest that GWP-042 may act as a powerful inhibitor of Rv2780.

*Mycobacterium marinum* (*M. marinum*), a pathogen of zebrafish that is the closest genetic relative of the *M. tuberculosis* organism complex[53], possesses a conserved homolog of alanine dehydrogenase (Rv2780) (Supplementary Fig. 3I). Zebrafishes have an antimicrobial peptide system[54] and have been used as a powerful host–pathogen system for characterizing anti-mycobacterial compounds[55,56]. From the top 15 compounds of the docking study with the best docking scores, GWP-042 was found to be the most effective inhibitor to restrict the growth of *M. marinum* in zebrafish larvae (Supplementary Fig. 9A, B),

but showed no significant effect on the growth rate of *M. marinum* in vitro (Supplementary Fig. 9C, D).

One hallmark of TB is the formation of caseous necrotic granulomas[57], which are organized aggregates of macrophages and other immune cells that serve as niches for the bacteria to obtain nutrients or evade anti-TB immunity, and to provide a source for mycobacteria for later reactivation and dissemination[58,59]. Respiration-inhibiting conditions, such as hypoxia, nitric oxide, low pH and nutrient starvation, are assumed to be characteristics of TB granulomatous lesions. Expression of the *ald* gene is upregulated under oxygen-limiting, nutrient starvation and nitrogen monoxide (NO) conditions[28,60–62]. The growth rate of *M. marinum* under hypoxia was not significantly affected by the treatment of GWP-042 (Supplementary Fig. 9D). Moreover, adult zebrafish treated with GWP-042 had a much lower bacterial burden of wild-type and rifampicin resistant *M. marinum* at 14 days post-infection (Supplementary Fig. 9E, F). These results suggest that targeting mycobacterial alanine dehydrogenase may inhibit the growth of pathogenic mycobacteria in granulomas.

In mice peritoneal macrophages infected with wild-type H37Rv, the addition of GWP-042 increased the production of Defb4 and Camp; but the increases were not observed upon infection with H37RvΔRv2780 strains (Fig. 6F and Supplementary Fig. 9G). These results suggest that GWP-042 may increase the AMPs by targeting Rv2780. However, GWP-042 had no significant effect on cytokines expression or NO production in *M. tuberculosis*-infected macrophages (Supplementary Fig. 10A, B). GWP-042 dramatically inhibited the intracellular survival of both *M. tuberculosis* H37Rv and a clinical MDR strain in infected macrophages (Supplementary Fig. 10C–F). However, treatment with GWP-042 showed no significant effect on the in vitro growth curve of *M. tuberculosis* H37Rv or MDR *M. tuberculosis* (Supplementary Fig. 10G–J), suggesting that GWP-042 may exert its anti-mycobacterial effect through targeting host anti-TB pathways. Moreover, the deletion of Rv2780 almost eliminated the inhibitory effect of GWP-042 on the growth of intracellular *M. tuberculosis* (Fig. 6G, H), indicating that GWP-042 may exert its anti-mycobacterial activity through inhibiting Rv2780. Furthermore, GWP-042 showed no significant effect on the viability of cells even at very high concentrations, no matter with or without H37Rv infection (Supplementary Fig. 10K, L). GWP-042 has no effect on the expression of cytochrome P450 (CYP450) homologs, *Cyp1a2*, *Cyp2b10*, *Cyp2c38*, *Cyp2d9* and *Cyp3a11*, in murine hepatocytes (Supplementary Fig. 10M), which suggests GWP-042 may not activate the CYP450 system. Together, these results suggest that targeting Rv2780 has potential as a host-directed candidate for the therapeutic treatment of TB, especially drug-resistant TB.

We further evaluated the pharmacokinetic properties of GWP-042. As shown in Supplementary Data 7, the half-life of GWP-042 was 2.07 and 2.25 h when C57BL/6J mice were treated by intravenous injection (10 mg/kg) and intragastric administration (100 mg/kg), respectively. We also observed a high maximal concentration ($C_{max}$ = 7237 ng/mL for intravenous injection and $C_{max}$ = 45425 ng/mL for intragastric administration) and a good bioavailability of 80.79% when GWP-042 was given orally. A clearance of 8.68 mL/min/kg

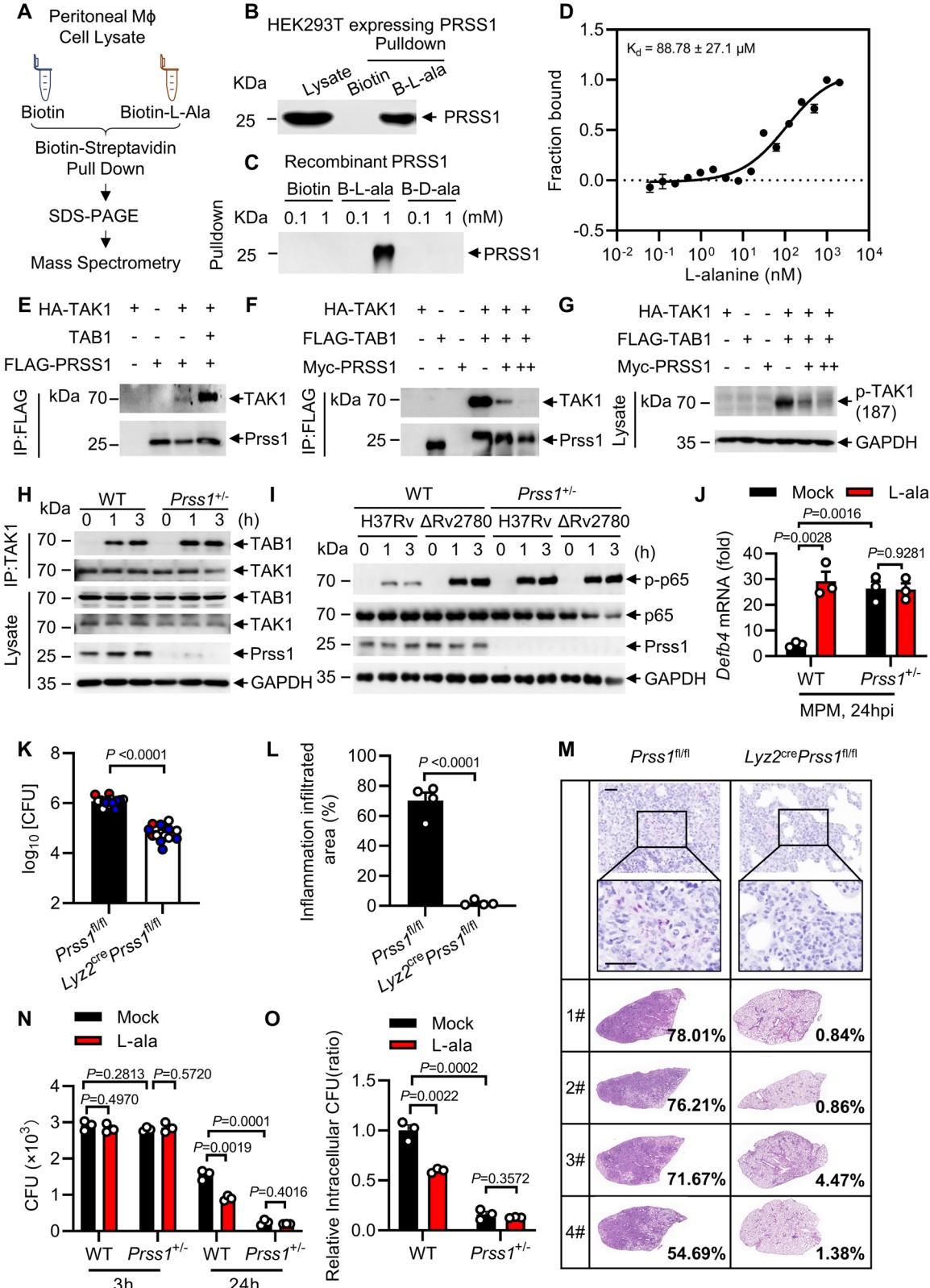

suggests the metabolic stability of GWP-042 was good. An in vivo toxicity study of GWP-042 was performed in C57BL/6J mice (Supplementary Data 8). No mice died after receiving 50 or 200 mg/kg by intragastric administration. When the dosage was raised up to 1000 mg/kg, two of three mice died. No significant change in body-weight was observed for C57BL/6J mice administrated with GWP-042 at 50 mg/kg once by oral gavage for 14 days (Supplementary Fig. 10N),

indicating that GWP-042 was nontoxic. Furthermore, when treated with GWP-042, the lung tissues of C57BL/6J mice infected with H37Rv had much lower bacteria burden and less inflammatory infiltration than those mice treated with rifampicin (Fig. 6I–K), indicating that the killing effect of GWP-042 against *M. tuberculosis* alone is better than rifampicin. These results suggest that targeting mycobacterial alanine dehydrogenase may inhibit the growth of pathogenic mycobacteria

**Fig. 4 | L-alanine interacts with Prss1 to induce NF/κB mediated *Defb4* expression. A** Work flow of biotin-streptavidin pulldown assay combined with mass spectrometry. Streptavidin pulldown assays of the binding of biotin-conjugated L-alanine to Flag-PRSS1 in HEK293T cells (**B**) or recombinant PRSS1 (**C**). Biotin-conjugated D-alanine was used as control (**C**). **D** MicroScale Thermophoresis (MST) assay of the direct interaction of L-alanine with Prss1. **E−G** Immunoblot and immunoprecipitation of HEK293T cells transfected with indicated plasmids. **H** Endogenous immunoprecipitation analysis of wild type (WT) or heterozygous *Prss1* knockout (*Prss1*[+/−]) MPMs infected with H37Rv for 0, 1 and 3 h (MOI = 2). **I** Immunoblot analysis of WT or *Prss1*[+/−] MPMs infected with H37Rv or H37RvΔRv2780 for 0, 1, and 3 h (MOI = 2). **J** RT-PCR analysis of *Defb4* in WT or *Prss1*[+/−] MPMs treated with 1mM L-alanine followed by H37Rv infection for 24 h (MOI = 2). CFU (**K**), histological score (**L**) and histopathological images (**M**) by acid-fast staining and haematoxylin and eosin staining in lung tissues of macrophage conditional *Prss1* knockout mice (*Lyz2*[cre]*Prss1*[fl/fl]) and control mice (*Prss1*[fl/fl]) after 28 days of H37Rv infection. Scale bars, 1000 μm (top; original magnification, ×40) and 200 μm (bottom; original magnification, ×100). CFU counts (**N**) and relative intracellular CFU ratio (**O**) in WT or *Prss1*[+/−] MPMs treated with 1 mM L-alanine followed by H37Rv infection for 3 and 24 h (MOI = 2). Data are representative of one experiment with at least three independent biological replicates; (**D, J, N, O**) *n* = 3 samples (mean ± s.e.m), each circle represents one technical repeat in (**J** and **N, O**); (**K**) *n* = 16 mice infected for 28 days with red, blue and white circles denoting separate experiments (mean ± s.e.m); (**L**) *n* = 4 mice (mean ± s.e.m). Two-tailed unpaired Student's *t*-test (**J, N, O**) and two-sided Mann-Whitney *U*-test (**K, L**) were used for statistical analysis. *P* values are shown in (**J, L** and **N, O**). 1#, 2# and 3# in (**M**) represent lung tissues from 3 mice. Scale bars, 100 μm (top; original magnification, ×400) and 20 μm (bottom; original magnification, ×1000). Source data are provided as a Source Data file.

in vivo. This is consistent with our conjecture that the mechanism of GWP-042 activity differs from that of traditional anti-TB drugs, which directly target *M. tuberculosis* itself; GWP-042 may resuscitate host immunity to eliminate *M. tuberculosis*.

## Discussion

Antimicrobial peptides are major components of host immunity, but previous studies have shown that they are very poorly induced in macrophages infected by *M. tuberculosis*[18–21]. Our findings identify the mycobacterial alanine dehydrogenase, Rv2780, as a previously unrecognized component of *M. tuberculosis* that suppresses the expression of AMPs. We found that PRSS1, a pancreatitis-associated factor[41,63], inhibits NF-κB−mediated expression of AMPs by disrupting the formation of the TAK1-TAB1 complex. L-alanine directly interacted with PRSS1, which disabled the latter's inhibitory effect on the TAK1/TAB1 complex formation, thereby triggering the NF-κB-mediated expression of AMPs. Nevertheless, *M. tuberculosis* secretes an alanine dehydrogenase Rv2780 that hydrolyzes L-alanine in host macrophages, thus suppressing the production of AMPs to facilitate the intracellular survival of mycobacteria. Thus, Rv2780 appears to be a virulence factor that allows *M. tuberculosis* to consume host metabolite L-alanine to evade host innate immunity. This mechanism depends on an ancient bactericidal mechanism, AMPs production, highlighting the versatility of host-*M. tuberculosis* interactions.

*M. tuberculosis* infection activates TLR2/NF-κB signaling pathway to induce the expression of AMPs[11], but how the induction of AMPs is negatively regulated remains unexplored. We found that heterozygous deletion of PRSS1, a pancreatitis-associated factor[41,63], markedly increases the expression of AMPs, indicating PRSS1 is a strong suppressor of AMPs expression. Moreover, our in vitro study showed that *Prss1*[+/−] mice peritoneal macrophages infected with *M. tuberculosis* H37Rv had much lower intracellular CFU than those WT counterparts. Given that antimicrobial peptides (AMPs) directly target intracellular bacteria, *Prss1* deficiency may mediate bacterial clearance in vitro mainly through regulating the expression of AMPs. In vivo study showed that significantly decreased lung bacterial burden and tissues damages were observed in *Lyz2*[cre]*Prss1*[floxp/floxp] mice infected with *M. tuberculosis*, suggesting that Prss1 may negatively regulate anti-TB immunity through downregulating the activation of NF-κB signal and not only AMP-related. Because higher NF-κB activation may not only induce AMPs expression, but also promote cytokines and chemokines expression in *Prss1* deficient macrophages, which may subsequently activate other immune cells (including neutrophils or T cells) to maintain the in vivo anti-TB immunity. Therefore, we could not exclude the involvement of other NF-κB regulated genes, which needs further exploration.

Mechanistically, PRSS1 interacted with TAK1 and disrupted the formation of TAK1-TAB1 complex to inhibit TAK1-mediated activation of NF-κB pathway, thus suppressing the expression of AMPs. Besides, NF-κB plays a central role in host response to different infection of pathogens[64,65], including gram-negative bacteria *E. coli*, gram-positive bacteria *S. aureus* and *M. tuberculosis*. Thus, the inhibitory effects of Prss1 on NF-κB activation were also observed during other bacterial infections. Considering multiple functions of AMPs[9,66–68], inhibition of AMPs by PRSS1 may keep the expression of AMPs in a quiescent state to avoid unnecessary side-effects. To the best of our knowledge, our study is the first time to show PRSS1 is an immune molecule that negatively regulates the expression of AMPs and anti-TB immunity. However, whether and how the protease activity of PRSS1 is involved in its suppression of AMPs await further investigation.

Pro-inflammatory cytokines were upregulated by Rv2780 in macrophages or lung tissue of *M. tuberculosis*-infected mice in spite of Rv2780 mediated inhibition of NF-kB activation. Although the expression of pro-inflammatory cytokines is mainly regulated by NF-κB activation, there are many NF-κB-independent mechanisms regulating cytokines expression such as epigenetic regulation of microRNA[69,70] or histone modification[71]. Besides, transcription factor Nrf2 suppresses inflammation through redox control without affecting NF-κB activation[72]. Thus Rv2780 may elevate inflammation cytokines expression through other NF-κB-independent pathways, which needs further investigation.

Amino acid metabolism has been shown to regulate immune responses to *M. tuberculosis* infection[73]. L-arginine is essential for macrophages to generate NO through inducible nitric oxide synthase[74]. *M. tuberculosis* requires the tryptophan biosynthetic pathway for their survival[75], but interferon-γ induces an isoform of the host enzyme, indoleamine 2,3-dioxygenase, that converts tryptophan to N-formylkynurenine, thus depleting tryptophan to exert an antimicrobial effect[76]. Alanine, an aliphatic neutral non-essential amino acid, is a critical structural component of mycobacterial cell wall peptidoglycan, and is utilized as a nitrogen source for the growth of *M. tuberculosis*[77,78], but its role in the regulation of immune responses to *M. tuberculosis* infection remains unclear. Our results indicate L-alanine is a strong inducer of AMPs that relieve the inhibitory effect of PRSS1, a pancreatitis-associated factor[41,63], by triggering NF-κB-mediated expression of AMPs. Furthermore, supplementation of alanine reduces the histopathologic damage and bacterial burden in the lung tissues of mice infected with *M. tuberculosis* H37Rv, indicating the therapeutic effectiveness of L-alanine in vivo. However, further study is needed to investigate whether alanine inhibits the protease activity of PRSS1 and confirm the anti-TB efficacy of L-alanine in nonhuman primates or humans.

*M. tuberculosis* infection reprograms host metabolism to exploit host metabolites for nutrients or to regulate host immunometabolism[73,79–83]. Ald was originally identified as one of the major antigens present in culture filtrates of *M. tuberculosis*[36]. It has a homohexameric quaternary structure with N-terminal catalytic and C-terminal NAD(H)-binding domains[26,27,35] and catalyzes the reversible conversion of L-alanine to pyruvate with concomitant reduction of NAD+ to NADH. The expression of the mycobacterial *ald* gene was strongly upregulated by alanine, nutrient starvation, hypoxia or in

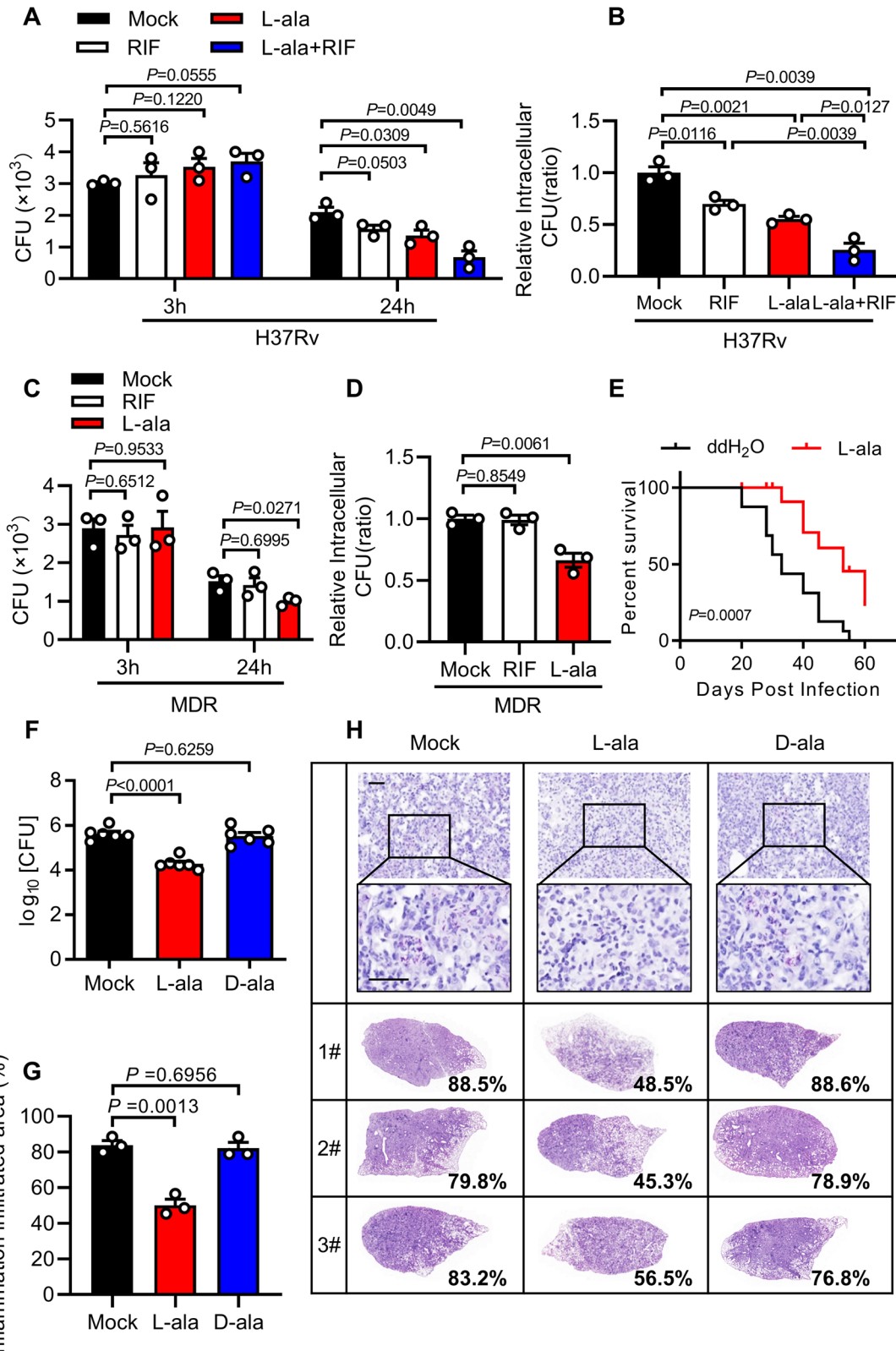

granuloma[60–62,84,85]. *Mycobacterium smegmatis* (*M. smegmatis*) Ald is required for utilization of alanine as a nitrogen source[29], and *M. bovis* BCG is unable to catabolize L-alanine due to a frameshift mutation in the *ald* gene[86]. In addition, Ald maintains the optimal NADH/NAD+ ratio for mycobacterial survival under respiration-inhibitory conditions and for reactivation when oxygen is enough for the regrowth of mycobacteria[29,61]. Other studies also reported that treatment of *M.*

*smegmatis* with bedaquiline, which inhibits the $F_1F_o$-ATP synthase by binding to c subunits, leading to the induction of *ald* expression[87]. We found that Rv2780, an *M. tuberculosis* Ald, dehydrogenates alanine inside infected macrophages to reduce alanine level and promotes the intracellular survival of *M. tuberculosis*, indicating that pathogenic mycobacteria may secret alanine dehydrogenase such as Rv2780 to suppress the expression of AMPs.

**Fig. 5 | Supplementation of L-alanine enhances anti-TB immunity.** CFU counts (**A**) and relative intracellular CFU ratio (**B**) in MPMs treated with 1 mM L-alanine, 50 nM rifampicin (RIF) or combination of 1 mM L-alanine and 50 nM RIF followed by H37Rv infection for 3 and 24 h (MOI = 2). CFU counts (**C**) and relative intracellular CFU ratio (**D**) in MPMs treated with 1 mM L-alanine or 50 nM rifampicin (RIF) followed by multidrug-resistant *M. tuberculosis* (MDR) infection for 3 and 24 h (MOI = 2). **E** Survival curve of 6-week-old female SCID mice treated with ddH$_2$O or 30 mg/ml L-alanine and aerosol infected with roughly 100 CFUs per mouse of H37Rv. CFU assay (**F**), histological score (**G**) and histopathological assay (**H**) with acid-fast staining and haematoxylin and eosin staining in lung tissues of C57BL/6J mice treated with ddH$_2$O or 30 mg/ml L-alanine after 28 days of H37Rv infection. Data in (**A**–**D**) are representative of one experiment with at least three independent biological replicates; (**A**–**D**) $n$ = 3, each circle represents one technical repeat (mean ± s.e.m); (**F**) $n$ = 6 mice (mean ± s.e.m); (**G**) $n$ = 3 mice (mean ± s.e.m). Two-tailed unpaired Student's $t$-test (**A**–**D**), Log-rank test (**E**) and two-sided Mann-Whitney $U$-test (**F**, **G**) were used for statistical analysis. $P$ values are shown in (**A**–**G**). 1#, 2# and 3# in (**H**) represent lung tissues from 3 mice. Scale bars, 100 μm (top; original magnification, ×400) and 20 μm (bottom; original magnification, ×1,000). Source data are provided as a Source Data file.

Several inhibitors of Ald have been developed[52,88–91], and their potent anti-TB activity has been validated in vitro. We found that the Ald-targeting inhibitor, GWP-042, restored the production of AMPs and inhibited the growth of mycobacteria in vivo. Notably, GWP-042 also inhibited the growth of pathogenic mycobacteria in a zebrafish granuloma model. As natural antimicrobial agents, AMPs have clear advantages over conventional antibiotics, including rapid and broad-spectrum bactericidal activity and limited emergence of resistance[92,93]. Considering the rapid growth and global spread of drug-resistant mycobacteria, targeting a mycobacterial immune evasion factor to boost primitive antimicrobial immunity may provide a window for the development of effective immunomodulators against TB, especially drug-resistant TB (Supplementary Fig. 11). Desjardins et al. hypothesized that strains lacking functional Rv2780 are unable to convert L-alanine to pyruvate, thereby increasing the pool of available L-alanine in *M. tuberculosis*. As L-alanine is the precursor to the pathway competitively inhibited by D-cycloserine, abundant L-alanine may allow for continued peptidoglycan production despite competitive inhibition by D-cycloserine[94]. Thus, Rv2780 inhibitor GWP-042 could be applied for other drug-resistant *M. tuberculosis* infection, except for D-cycloserine-resistant strains. Furthermore, given that the expression of the *ald* gene is induced under respiration-inhibitory conditions in *M. smegmatis* and inactivation of the *ald* gene exacerbates the growth defect of *M. smegmatis* by respiration inhibition[29], the combination of GWP-042 with respiration-inhibitory anti-TB drugs, such as Q203[95] and bedaquiline[87], are likely to be a more efficient treatment against TB, especially latent TB, which currently lacks an efficient drug target. However, the anti-TB efficacy of the Ald-targeting inhibitor, GWP-042, in nonhuman primates or humans needs further investigation.

In summary, our findings propose an intriguing model for the regulation of AMPs expression by host and pathogen interaction: first, PRSS1 acts as a negative regulator to keep the expression of AMPs in a quiescent state to avoid unnecessary side effects; in response to mycobacteria infection, L-alanine is induced to disable PRSS1-mediated inhibition, thus triggering AMPs expression for the clearance of bacteria; however, pathogenic mycobacteria have evolved an alanine dehydrogenase that suppresses the host AMPs through dehydrogenating L-alanine. Considering AMP's efficient and broad-spectrum bactericidal activity but limited emergence of resistance, targeting the mycobacterial virulence factor to boost the host AMPs may provide a window for the development of effective immunomodulators against TB, especially for drug-resistant TB.

## Methods
### Relevant ethical regulations
All protocols were approved by the local ethics committee of Shanghai Pulmonary Hospital (permit number: K23-333Z) or the local ethics committee of Tongji University (permit number: TJAA06522101). This study was conducted according to the Declaration of Helsinki principles and signed informed consent was obtained from all subjects. Specifically, the collection and use of samples from TB patients were approved by the local ethics committee of Shanghai Pulmonary

Hospital (permit number: K23-333Z). The use of animal in our work was approved by the local ethics committee of Tongji University (permit number: TJAA06522101).

### Bacterial culture and infections
*Mycobacteria tuberculosis* (*M. tuberculosis*) H37Rv strains (Supplementary Data 9) were grown in Middlebrook 7H9 broth (7H9, BD Biosciences) supplemented with 10% oleic acid-albumin-dextrose-catalase (OADC), 0.5% glycerol (Sigma-Aldrich) and 0.05% Tween-80 (Sigma-Aldrich), or on Middlebrook 7H10 agar (BD Biosciences) supplemented with 10% OADC.

H37RvΔRv2780 was constructed by Shanghai Gene-optimal Science & Technology Co., Ltd. according to previous publications[96,97]. In brief, screening gene cassette sacB-hygromycin B was inserted H37Rv genome replacing Rv2780 gene through recombinant phage with homologous gene by homologous recombination. H37RvΔRv2780 was confirmed by PCR and western blot. The shuttle vector pMV261 (provided by K. Mi, Institute of Microbiology, Beijing, China) was used to complement the strain H37RvΔRv2780 with wild-type Rv2780 (H37RvΔRv2780 + Rv2780) or to create the strain H37RvΔRv2780 + Rv2780$^{DM}$. Expression of Rv2780 or its mutants (with a C-terminal Flag-tag) in mycobacteria was examined by immunoblot analysis. For H37RvΔRv2780, 50 μg/ml hygromycin B was added to culture. For H37Rv (ΔRv2780 + Rv2780) or H37Rv (ΔRv2780 + Rv2780$^{DM}$), 50 μg/ml hygromycin B and kanamycin were added to culture. *E. coli* DH5a/BL21 or *S. aureus* were grown in LB medium.

For macrophages infection, mice peritoneal macrophages or BMDMs were seeded in six-well plates (1 × 10$^6$ cells/well) and cultured for 24 h at 37 °C in a 5% CO$_2$ incubator. *M. tuberculosis*, *E. coli* DH5a/BL21 or *S. aureus* were added to cells at MOI = 2.

### Cell culture
HEK293T cells (ATCC CRL-3216), A549 (ATCC CRM-CCL-185) and AML-12 cells (ATCC CRL-2254) were obtained from the American type culture collection (ATCC). The cells were cultured in Dulbecco's modified Eagle's medium (DMEM, HyClone) supplemented with 10% (v/v) heat-inactivated fetal bovine serum (FBS, Gibco) and 100 U/ml penicillin and streptomycin. The transient transfection of HEK293T was carried out using polyethylenimine (Polysciences) or Lipofectamine 2000 (Invitrogen). All the cells were routinely tested for mycoplasma contamination, and only those tested negative cells were used for experiments.

Mice peritoneal macrophages were harvested from mice that were injected 10% thioglycollate (BD Biosciences) for 3 days. The peritoneal macrophages were cultures in RPMI-1640 medium (HyClone) supplemented with or without 10% (v/v) FBS. BMDMs were obtained from isolated mouse bone marrow cells followed by incubation in 10% (v/v) FBS, 40 ng/ml M-CSF (Peprotech) and 20 ng/ml IL-4 (Peprotech) for 7 days. Mice alveolar macrophages were generated from single-cell suspensions of bronchial alveolar lavage fluid via macrophage adherence[98,99]. Mice bone marrow-derived neutrophils were isolated from mouse bone marrow cells via EasySep™ Mouse Neutrophil Enrichment Kit (Stemcell).

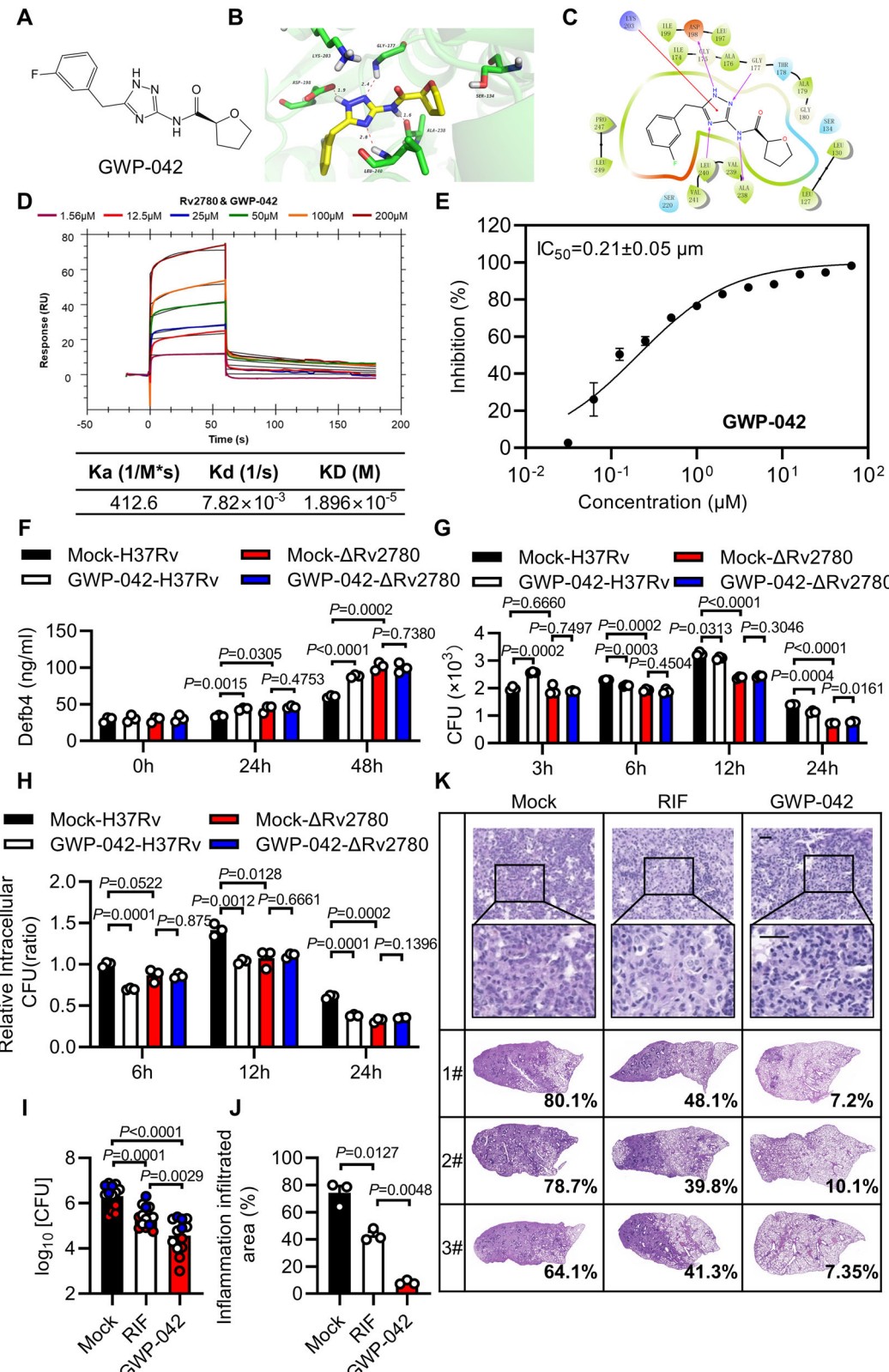

## Plasmids, antibodies, and reagents

Plasmids are described in Supplementary Data 1 and Supplementary Data 7. The polyclonal rabbit anti-Rv2780 antibody was produced and purified by ABclonal Biotech. The following antibodies were used in this study: rabbit anti-HA antibody (H6908/polyclonal, Sigma-Aldrich, 1:2000 for immunoblot analysis); rabbit anti-GAPDH antibody (SAB2701826/polyclonal, Sigma-Aldrich, 1:2000 for immunoblot

analysis), rabbit anti-FLAG antibody (F7425, Sigma-Aldrich, 1:2000 for immunoblot analysis), Goat anti-Rabbit IgG (H + L) Secondary Antibody, Alexa Fluor 488 (A-11008, Invitrogen, 1:500 for immunofluorescence), anti-FLAG M2 Magnetic Beads (M8823, Sigma-Aldrich, for immunoprecipitation), rabbit Anti-PRSS1 antibody (ab200996/ monoclonal, Abcam, 1:1000 for immunoblot analysis), rabbit anti-NF-κB p65 (C22B4) antibody (4764/monoclonal, Cell Signaling

**Fig. 6 | Targeting L-alanine pathway enhances anti-TB immunity. A** Structure of GWP-042. Superimposition image (**B**) of docked pose of the GWP-042 to Rv2780 protein. Ligand-target interaction diagram (**C**) showing interactions of GWP-042 with the active site residues of Rv2780 protein. **D** Surface plasmon resonance (SPR) assay of the direct interaction of GWP-042 with Rv2780 protein. (**E**) The dehydrogenase activity of Rv2780 was measured by pyruvate concentration in the presence of increasing concentrations of GWP-042. Dose−response curves for $IC_{50}$ values were determined by nonlinear regression. **F** ELISA analysis of Defb4 in MPMs treated with 50 μM GWP-042 followed by H37Rv or H37RvΔRv2780 infection for 0, 24 and 48 h (MOI = 2). CFU counts (**G**) and relative intracellular CFU ratio (**H**) in mice peritoneal macrophages treated with 50 μM GWP-042 followed by H37Rv or H37RvΔRv2780 infection for 3, 6, 12, and 24 h (MOI = 2). CFU assay (**I**), histological score (**J**) and histopathological assay with acid-fast staining and haematoxylin and eosin staining (**K**) in lung tissues of C57BL/6 J mice aerosol-infected with roughly 200 CFUs per mouse of H37Rv for 21 days, and treated with Rifampicin (RIF) (10 mg/kg/day) or GWP-042 (10 mg/kg/day) via oral gavage for another 28 days. Data in (**E**−**J**) are representative of one experiment with at least three independent biological replicates; (**E**−**H**) $n$ = 3 samples (mean ± s.e.m), each circle represents one technical repeat in (**F**−**H**); (**I**) $n$ = 13 mice infected for 28 days with red, blue and white circles denoting separate experiments (mean ± s.e.m); (**J**) $n$ = 3 mice (mean ± s.e.m). Two-tailed unpaired Student's $t$-test (**F**−**H**) and two-sided Mann-Whitney $U$-test (**I**, **J**) were used for statistical analysis. $P$ values are shown in (**F**−**J**). 1#, 2# and 3# in (**K**) represent lung tissues from three mice. Scale bars, 100 μm (top; original magnification, ×400) and 20 μm (bottom; original magnification, ×1000). Source data are provided as a Source Data file.

Technology, 1:1000 for immunoblot analysis), rabbit anti-TAK1 (D94D7)antibody(5206/monoclonal, Cell Signaling Technology, 1:1000 for immunoblot analysis, 1:50 for immunoprecipitation), rabbit anti-TAB1 antibody (A5749/polyclonal, Abclonal, 1:1000 for immunoblot analysis), rabbit anti-phospho-TAK1 (Thr187) antibody (4536/polyclonal, Cell Signaling Technology, 1:1000 for immunoblot analysis), rabbit anti-phospho-NF-κB p65 (Ser536) antibody(3033/monoclonal, Cell Signaling Technology, 1:1000 for immunoblot analysis), purified anti-*E. coli* RNA Sigma 70 antibody(663208, BioLegend, 1:1000 for immunoblot analysis of RpoD). GWP-042 was purchased from MedChemExpress (Cat.No.: HY-45854), the purity of the compound was 95.63% as detected by the company.

## Immunofluorescence and confocal microscopy

Mice peritoneal macrophages were plated in Glass Bottom Culture Dishes (NEST, 801002) and infected with *Mycobacterium tuberculosis* H37Rv (MOI = 2) for 24 h. After infection, cells were stained with MitoTracker Deep Red FM (Invitrogen, M22426), LysoTracker Red DND-99 (Invitrogen, L7528) or ER-Tracker Red (Beyotime, C1041S) at 37 °C for 30 min in the dark. Cells were washed with precooled PBS for three times, 5 min each time, and were then fixed, permeabilized, and blocked at room temperature. Primary anti-Rv2780 antibodies (Abclonal, customized) were then applied at 4 °C overnight. After washing three times with PBS, culture dishes were incubated with Alexa Fluor 488-conjugated secondary antibodies for 1 h followed by staining with DAPI. For autophagic flux analysis, primary peritoneal macrophages were pre-treated with adenovirus expressing mCherry-GFP-LC3B fusion protein for 48 h before H37Rv or ΔRv2780 infection. Confocal images were taken with the Leica SP8 confocal microscope (Leica Microsystems) and analyzed by the Leica Application Suite Las X (v2.0.1.14392) software.

## Mice and Infection

*Prss1*[+/-] mice on C57BL/6J genetic background were purchased from the Cyagen Biosciences. Macrophage conditional knockout mice female 6-8 weeks old SPF C57BL/6J and SCID mice were purchased from Slaccas for peritoneal macrophages and BMDMs separation. *Prss1*[floxp/floxp] mice and *Lyz2*[cre] mice on C57BL/6J genetic background were purchased from Shanghai Model Organisms Center. Macrophage conditional *Prss1* knockout mice were generated by breeding *Prss1*[floxp/floxp] mice and *Lyz2*[cre] mice.

All the mice infection experiments were performed with age- and sex-matched groups of 8–12-weeks old mice. Each mouse was aerosol infected with H37Rv, H37RvΔRv2780, H37Rv(ΔRv2780 + Rv2780) or H37Rv(ΔRv2780 + Rv2780[DM]) (100–200 CFUs) for 0, 14 and 28 days. The pathological sections were examined with hematoxylin and eosin (H&E) stain and bacteria load in lung section was analyze by acid-fast staining. The digital images were captured by 3Dhistech Pannoramic Scan system (3DHISTECH Ltd.) and processed by CaseViewer™ application. The histopathological scores were calculated by dividing inflammatory infiltrated area by total lung area. For CFU analysis, lung tissues were homogenized and plated on 7H10 agar plates for CFU counting. For L-alanine supplement mice infection experiments, C57BL/6 J mice drink double distilled water or 30 mg/ml L-alanine (Sangon Biotech) two days before infection and lasted until the end of the experiment. To rule out the effect of L-alanine on regulating adaptive immune cell responses, SICD mice, which are adaptive immune deficient, were used examine the functional role of L-alanine in regulating innate immunity in vivo by survival analysis of *M. tuberculosis*-infected mice. To validate the effectiveness of tuberculosis drug regimens, C57BL/6 J mice were treated with RIF (Rifampicin) (10 mg/kg/day), or combination of RIF (10 mg/kg/day) and Rv2780 inhibitor GWP-042 (MedChemExpress, 91%) (10 mg/kg/day) via oral gavage after three weeks of H37Rv aerosol infection. After 4 weeks of treatment, all mice were sacrificed for analysis of lung bacterial burden and histopathology. All animal experiments were reviewed and approved by the Animal Experiment Administration Committee of Shanghai Pulmonary Hospital.

## Zebrafish and infection

According to the reported assay, 300 CFU of Td-Tomato labeled *Mycobacterium marinum* (Supplementary Data 7) were injected into caudal vein of zebrafish larva at 48 hpf (hours post fertilization). Zebrafish larva were transferred to drug-containing egg water since 2 days post infection. Overall bacterial burden of whole larvae can be quantified at 5 days post infection by fluorescence microscopy followed by fluorescence quantification of images via imageJ. Adult zebrafish (EzeRinka Biotech) were infected with 200 CFU of Td-Tomato labeled *Mycobacterium marinum* by intraperitoneal injection as previously reported[71]. Drug treatment(4 mg/kg) was conducted once a day via oral gavage after 7 days post infection. To access the bacterial burden, zebrafishes infected for 14 days were euthanized in Tricaine (Sigma), followed by plating whole body homogenate on 7H10 agar plates.

## Intracellular CFU assay

MPMs were infected with H37Rv (MOI = 2) or other H37Rv strains. Cells were incubated for 3 h at 37 °C with 5% $CO_2$. Subsequently, cells were washed three times with sterile PBS, and were then incubated in fresh RPMI medium. CFUs were enumerated at indicated time. For CFU enumeration in infected cells, supernatants were removed and cell pellets were lysed with sterile 1% Triton followed by gradient dilutions in PBS. Serial dilutions in PBS were plated on 7H10 agar plates with 10% OADC enrichment. Plates were then incubated at 37 °C and counted after 21 days. The relative intracellular CFU ratio was calculated through dividing CFU counts at the corresponding time by CFU counts at 3 h.

## Quantitative metabolomic analysis

Sera from H37Rv-infected mice and age-matched control were processed at Biotree, Inc. (Shanghai, China) by a gas chromatograph coupled with a time-of-flight mass spectrometer (GC-TOF-MS, Agilent

7890) using a DB-5MS capillary column. Helium was used as the carrier gas, the front inlet purge flow was 3 mL/min, and the gas flow rate through the column was 1 mL/min. The initial temperature was kept at 50 °C for 1 min, then raised to 310 °C at a rate of 20 °C/min, then kept for 6 min at 310 °C. The injection, transfer line, and ion source temperatures were 280, 280 and 250 °C, respectively. The energy was −70 eV in electron impact mode. The mass spectrometry data were acquired in full-scan mode with the m/z range of 50–500 at a rate of 12.5 spectra per second after a solvent delay of 4.8 min. Raw data analysis, including peak extraction, baseline adjustment, deconvolution, alignment and integration, was finished with Chroma TOF (V 4.3×, LECO) software and LECO-Fiehn Rtx5 database was used for metabolite identification by matching the mass spectrum and retention index.

## Carbon flux analysis with U13C glucose

MPMs were seeded on 90 mm Petri dishes ($1 \times 10^7$ cells per dish) in RPMI 1640 medium. One hour before the infection, the culture medium was replaced by RPMI 1640 containing 5 mM U13C glucose without FBS followed by infection with indicated strains for 24 h. Cells were scraped in 80% methanol and phase separation was achieved by centrifugation at 4 °C and the methanol-water phase containing polar metabolites was separated and dried using a vacuum concentrator. The dried metabolite samples were derivatized for GC/MS analysis as follows: First, 70 μl of O-Isobutylhydroxylamine hydrochloride was added to the dried pellet and incubated for 20 min at 85 °C. After cooling, 30 μl of N-tert-butyldimethylsilyl-N-methyltrifluoroacetamide (MTBSTFA) was added and samples were re-incubated for 60 min at 85 °C before centrifugation for 15 min at $13,400 \times g$ (4 °C). The supernatant was transferred to an autosampler vial for GC/MS analysis. Isotopologue distributions and metabolite levels were measured with a Shimadzu QP-2020 GC-MS system.

GC/MS data were analyzed to determine isotope labeling and quantities of metabolites. To determine 13C labeling, the mass distribution for known fragments of metabolites was extracted from the appropriate chromatographic peak. These fragments contained either the whole carbon skeleton of the metabolite, or lacked the alpha carboxyl carbon, or (for some amino acids) contained only the backbone minus the side-chain[100]. For each fragment, the retrieved data comprised mass intensities for the lightest isotopomer (without any heavy isotopes, M0), and isotopomers with increasing unit mass (up to M6) relative to M0. M + 0 to M + n indicate the different mass isotopologues for a given metabolite with n carbons, where mass increases due to 13C-labeling.

## ELISA of antimicrobial peptides

ELISA analysis for protein level of Camp and Defb4 was conducted as previous reported[37,38]. Cell pellets of MPMs infected with H37Rv strains for indicated time were harvested and lysed in the lysis buffer lysed with RIPA Lysis Buffer (Beyotime). Cell lysates and supernatants were collected for ELISA analysis. According to the manufacturer's instructions, Camp levels were measured by ELISA kit (Cusabio Biotech Co., Ltd.), while Defb4 levels were measured by ELISA kit (Fine Biotech Co., Ltd.).

## Alanine detection assay

Alanine level of cellular and tissue samples was tested by Alanine Assay Kit (Sigma-Aldrich Co. LLC., MAK001). Cell pellets of MPMs infected with H37Rv strains were lysed in Alanine Assay Buffer and centrifuged at 4 °C for 10 min at 12,000 g, and the supernatants were used for alanine detection. Supernatants of lung homogenate and sera were diluted with Alanine Assay Buffer. According to the manufacturer's instructions, all samples were collected and deproteinized before use in assay with a 10 kDa Molecular Weight Cut-Off spin filter (Sigma-Aldrich Co. LLC.). The filtrates were detected for analysis of alanine amount, while the unfiltered concentrates were detected by BCA

protein assay for quantification protein level or immunoblotting analysis of GAPDH, an indicator that can reflect the number of cells. Then the normalization alanine level of macrophages was calculated as follows: Normalized alanine level = Alanine concentration/Gray value of GAPDH. The normalization alanine level of sera or lung homogenate was calculated as followings: Normalized alanine level = Alanine concentration / BCA protein concentration.

## Virtual screening of Rv2780 inhibitors

For virtual screening of Rv2780 inhibitors, 2D structures of 309800 compounds from Hit Locator Library 300 and MCE Bioactive Compound Library and 3D structure of MtAlaDH (PDB ID: 2VHX) were submitted to Schrödinger' Glid docking module level (HTVS, SP, XP) for molecular docking. The compounds were ranked according to their binding affinity to target protein Rv2780. Top 50 compounds were selected for subsequent functional screening. Figures of molecular docking structures were generated using PyMOL (The PyMOL Molecular Graphics System, Schrödinger, LLC).

## Pharmacokinetics of GWP-042

Male SPF ICR mice were treated with a solution of GWP-042(DMSO/Solutol/Saline, 5/10/85, v/v/v/v) at a single dose of 10 mg/kg or a solution of GWP-042 dissolved in 0.5% CMC-Na at a single dose of 100 mg/kg via intravenous injection (iv) or oral administration(op), respectively. Blood samples were collected at 0.5, 1, 2, 4, 6, 8, 12, 24 h after administration. Plasma was separated and the concentration of compounds in plasma was calculated by LC-MS/MS analysis.

## Intracellular ROS production assays

Mice peritoneal macrophages were infected with H37Rv or as indicated and incubated with fresh medium added containing 10 μM DCFH-DA (Sigma-Aldrich) at 37 °C for 30 min. The cells were then washed with PBS 3 times and lysed with lysis buffer (50% methanol containing 0.1 M NaOH). After gently stripping the cells from the plate and spinning at $2671 \times g$ for 5 min, the supernatants were transferred and fluorescence at 488/525 nm was detected using a Synergy H1 multi-mode reader (Biotek). All the data were normalized with protein concentration.

## MTT assay

Mice peritoneal macrophages were seeded in 96-well plates. L-alanine and GWP-042 were added into the medium and plates were incubated for 24 h followed by MTT assays. For analysis during M. tuberculosis infection, MPMs treated with L-alanine or GWP-042 followed by infection with indicated strains for 0, 3, 6, 9, 12, 24 h. Cells were washed once with PBS and incubated in MTT Solvent containing 5 mg/mL MTT (3-(4,5-dimethylthiazol-2-yl) - 2,5-diphenyltetrazolium bromide) (Beyotime). Following 4 h incubation at 37 °C, cells were added with Formazan Solvent and mixed gently and continue incubation until Formazan is completely dissolved. Absorbance was measured at 570 nm using a TECAN microplate reader.

## In vivo toxicity of GWP-042

Female 8 weeks C57BL/6 J mice were administrated with GWP-042 dissolved in 0.5% CMC-Na at a single dose of 50 mg/kg, 200 mg/kg and 1000 mg/kg respectively by oral gavage for acute toxicity study. The mice were observed for mortality and toxic signs for 14 days.

## Western blot analysis and immunoprecipitation

For immunoblot analysis, cells were lysed in the 1× loading buffer (50 mM Tris-HCl(pH6.8), 2% SDS, 10%glycerol, 1% β- mercaptoethanol, 0.1% Bromophenol BLUE). After boiled for 10 min denaturation, Proteins were separated by SDS-PAGE and transferred to nitrocellulose filter membrane (Whatman), The membranes were blocked with 5% BSA in TBST buffer (0.1% Tris, 0.1% Tween-20) for 1 h at room

temperature and subsequently incubated with primary antibodies overnight at 4 °C. The membranes were washed three times with TBST before incubation with secondary antibody for 1 h at room temperature. After another three washes, analysis was performed using chemiluminescence reagent (Thermo Scientific).

For immunoprecipitation (IP), cells were lysed in Western blot and IP cell lysate buffer (20 mM Tris(pH7.5), 150 mM NaCl, 1% Triton X-100) supplemented with protease inhibitor cocktail (MedChemExpress). After centrifugation, supernatants were incubated with anti-FLAG M2 Magnetic Beads (Sigma-Aldrich) overnight at 4 °C. The samples were washed three times with PBST ($KH_2PO_4$ 2 mM, $Na_2HPO_4$ 8 mM, NaCl 136 mM, KCL 2.6 mM, 1% Triton X-100) and subjected to Western blot analysis. For biotin pull down assay, PRSS1 recombinant protein or cell lysates of HEK293T cells overexpressed with Flag-PRSS1 were incubated with biotinylated alanine or biotinylated serine for 2 h followed by incubation with Streptavidin Magnetic Beads (MedChemExpress) overnight at 4 °C. For competition assay, PRSS1 recombinant proteins were incubated with increasing concentrations of non-biotinylated alanine before incubation with biotinylated amino acids.

### Identification of H37Rv knockout and complementary strains
Various H37Rv strains were cultured to log phase culture and centrifugated at 12,000 $g$ for 10 min. For immunoblotting, bacterial pellets were washed three times with PBS buffer, and denatured at 95 °C with 1×SDS loading buffer (Tris-HCl pH 8.0 10 mM, DTT 50 mM, SDS 1%, Glycerol 10%, Bromophenol Blue 0.008%) for 10 min. Culture supernatants were mixed with adequately with an organic solvent (supernatant: methyl alcohol: chloroform = 4:4:1, v/v/v) followed by centrifugation at 4 °C for 10 min at 12,000 $g$ to isolate crude protein extract. Insoluble substances were denatured at 95 °C with 1×SDS loading buffer and subjected to Western blot analysis. Anti-RpoD antibody (BioLegend, 663208) was used as control.

### RT-PCR analysis
Transfected HEK293T cells or *M. tuberculosis* infected macrophages in 12-well plates were lysed by 1 ml TRIzol reagent (Invitrogen). Total RNA is precipitated from with chloroform and isopropanol according to the manufacturer's instructions. The first-strand complementary DNA (cDNA) was synthesized using the ReverTra Ace-α-First-Strand cDNA Synthesis Kit (Toyobo Biologics) according to the manufacturer's instructions. Lastly, qPCR analyses were performed with SYBR Grreen realtime PCR Master Mix (TOYOBO) on ABI 7300 system (Applied Biosystems) using gene-specific primers (Supplementary Data 7).

### Luciferase assay
HEK293T cells were transiently transfected with pNF-κB–luc, pRL–TK plasmids and the indicated plasmids for 24 h. The Dual-Luciferase reporter assay system (Promega, Madison, USA) was used for the detection of luciferase activity.

### In vitro Rv2780 enzyme assay
The reaction buffer was 125 mM glycine/KOH (pH 10.2), increasing concentration of L-alanine (0, 10, 100 mM), 1.25 mM $NAD^+$ and 6.026 pM of Rv2780 protein in a final volume of 200 μL. The reactions were carried out in 96-well plate at 37 °C. Inhibitors at indicated concentrated were incubated with Rv2780 protein before the reaction. The reaction was measured by the production of NADH via $NAD^+$/NADH Assay Kit with WST-8 (Beyotime). For direct detection of enzymatic product pyruvate, the reaction mixture was deproteinized with a 10 kDa MWCO spin filter (Sigma-Aldrich Co. LLC.) before quantification of pyruvate level with Pyruvate Assay Kit (Abcam, ab65342).

### Clinical samples
All the TB patients providing blood samples were from Shanghai Pulmonary Hospital between 2020 and 2021. They were diagnosed based on chest X-rays, acid-fast bacillus staining of biofluids samples, culture on Lowenstein–Jensen media and were corroborated with clinical symptoms. Patients were given informed consent. The ethics committee of Shanghai Pulmonary Hospital approved this consent procedure (permit number: K23-333Z). X-ray scores were calculated according to previously reported[101]. Lungs were divided into six zones (low, middle, and high zones for each left and right lung). The score was based on the percentage of lung parenchyma that showed evidence of each recorded abnormality: (l) involvement of less than 25% of the image; (2) 25% to 50%; (3) 50% to 75%; (4) more than75%. A profusion score (l to 4) was given and the scores of each zone were then summed to obtain a global profusion score for chest CT. Total weighted X-ray score is equal to score × 100/24 (total score) + 40 (if cavitation is present).

### Microscale thermophoresis
Purified human recombinant protein PRSS1 were labeled by Monolith NT Protein labeling kit RED−NHS (Nano Temper Technologies, Germany) according to the manufacturer's protocol. 10 μl of 40 nM labeled proteins were incubated with 10 μl of increasing concentrations of L-alanine (250−0.007 μM) in Assay Buffer (50 mM HEPES, pH7.5, 500 mM NaCl, 5% Glycerol, 1 mM TCEP). Then, samples were loaded into standard glass capillaries (Monolith NT Capillaries, Nano Temper Technologies) and the MST analysis was performed on a NanoTemper Monolith NT.115 apparatus (Nano Temper Technologies, Germany).

### Surface plasmon resonance (SPR)
The interaction of GWP-042 with Rv2780 was detected by OpenSPR™ (Nicoya Lifesciences, Waterloo, Canada). Briefly, Rv2780 protein was fixed on the COOH sensor chip by capture-coupling, then GWP-042 at indicated concentrations was injected sequentially into the chamber in PBS at 25 °C. The binding time was 240 s and the disassociation time was 360 s with the flow rate of 20 μl/min. The chip was regenerated with 10 mM Glycine-HCl with a flow rate of 150 μl/min. A one-to-one diffusion corrected model was fitted to the wavelength shifts corresponding to the varied glycan concentration. The kinetic constants, including the association constant ($k_a$), dissociation constant ($k_d$), and affinity (KD, KD = kd/ka), were analyzed with TraceDrawer software (Ridgeview Instruments AB, Sweden).

### Statistical analysis
Statistical significance between groups was determined by two-tailed Student's *t*-test, two-tailed analysis of variance followed by Bonferroni post hoc test, Log-rank test or two-sided Mann-Whitney *U*-test. Differences were significant at $P < 0.05$. The experiments were not randomized, and the investigators were not blinded to allocation during experiments and outcome assessment.

### Reporting summary
Further information on research design is available in the Nature Portfolio Reporting Summary linked to this article.

## Data availability
Source data are provided in this paper. *M. tuberculosis* secreted protein screening, metabolite profiling, carbon metabolic flux, mass spectrometry, preliminary pharmacokinetic evaluation and in vivo toxicity data in this study are available in Supplementary Data file. Further information and requests for resources or reagents should be directed to and will be fulfilled by Lin Wang (651377481@qq.com) or Baoxue Ge (gebaoxue@sibs.ac.cn). Source data are provided with this paper.

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

## Acknowledgements

We thank Prof. K. Mi (CAS Key Laboratory of Pathogenic Microbiology and Immunology) for the pMV261 plasmid and members of B. Ge's laboratory (Shanghai Key Laboratory of Tuberculosis, Shanghai Pulmonary Hospital, Tongji University School of Medicine, Shanghai, China) for helpful discussions and technical assistance. This project was supported by the National Key R&D Program of China (2023YFC2307300 and 2022YFC2302900 to L.W.; 2021YFA1300902 to R.Z.; 2017YFA0505900 to B.G.); National Natural Science Foundation of China (32188101, 32030038, 91842303, and 3170025 to B.G.; 82122029 and 82071776 to L.W.; 82100007 to P.W.); The Most Important Clinical Discipline in Shanghai (2017ZZ02003); Shanghai Rising-Star Program (20QA1408400 to L.W.); and the Shanghai "Chen Guang" project (19CG22 to L.W.). The fellowship of China National Postdoctoral Program for Innovative Talents (BX2021215 to Y.D.).

## Author contributions

Conceptualization: B.X.G., L.W. Methodology: C.P., Y.N.C., M.T.M., Q.C., Y.J.D., H.Y., Z.H.L.,. R.J.Z., J.X.C., J.W., X.C.H. Investigation: C.P., L.W., Y.N.C., M.T.M., Q.C., H.Y. Animal infection experiments: C.P., L.W., Y.N.C., M.T.M., H.Y., Y.J.D., S.S.L., H.Y.C., X.Y.W., Z.J.; Macrophages isolation, infection and Q-PCR.: C.P., J.P.H., W.Y.B., C.Y.S.; Western blot and confocal imaging: C.P., L.W., Y.N.C., Q.C.; Strains construction and preservation: J.W. and X.C.H.; Clinical samples and analysis: P.W. and W.S.; Writing: B.X.G., L.W., C.P.

## Competing interests

The authors declare no competing interests.
