## [Peer Review File · Nature Communications]

REVIEWER COMMENTS

Reviewer #1

Expertise - Host pathogen interactions, in vivo murine models

(Remarks to the Author):

The manuscript by Peng et al. "Pathogenic mycobacteria suppress host antimicrobial peptides by dehydrogenating L-alanine" reports that a mycobacterial (*M. tuberculosis*, Mtb) secreted protein, notably Rv2780, alters production of AMP in various cells by targeting L-alanine abundance. The authors propose that this amino acid in turn releases the inhibitory effects of the protease serin 1, PRSS1, on NF- κ B-driven AMP production. Supplementation with L-alanine and delivery of a compound which targets Rv2780 showed beneficial effects in the murine TB model. The conclusions are not fully supported by the data. Even the title is misleading because 1) Rv2780 is conserved in non-pathogenic mycobacteria such as *M. smegmatis* (Suppl. Fig. 3), 2) roles beyond AMP have not been tested, and 3) dehydrogenation of L-alanine has not been shown in the infection context. Detailed comments are presented below.

General comments:

1. Data suggesting a direct inhibition of AMP by Rv2780 are not convincing.

Firstly, alanine is a non-essential amino acid which can be synthesized from pyruvate. The author demonstrated that overexpression of Rv2780 indeed reduced the level of alanine in HEK293T and A549 cells. However, infection of macrophages with Mtb and Mtb mutants did not result in significant difference of alanine concentration (Fig. 2G, 2H). Therefore, it is questionable whether the amount of Rv2780 during infection can lead to significant reduction of alanine. Levels of Rv2780 at 3 and 6 hpi in MPM lysates are provided and seem impressive (Suppl. Fig. 1). yet it remains questionable how Mtb can produce high amounts of likely secreted protein in such a short time.

Secondly, according to Fig. 2A, Rv2780 catalyzes interconversion of alanine and pyruvate. It is also possible that Rv2780 decreases the level of pyruvate. The authors should perform metabolite profiling of cells treated with Rv2780.

Thirdly, it is likely that Rv2780 directly inhibits NF- κ B activation. The author should check this possibility. If so, why only AMP levels were tested? Why pro-inflammatory cytokines, which are relevant for anti-mycobacterial responses, remained not investigated? All mouse studies are performed at adaptive stages, which makes it difficult to disentangle AMP roles from complex host factors affecting TB susceptibility.

Lastly, the non-pathogenic mycobacterial species, including *M. smegmatis* (Suppl. Fig. 3) also have the homologue of Rv2780. Do they show effects similar to Mtb Rv 2780?

2. The interaction between PRSS1 and alanine is arguable.

Is PRSS1 the only hit in the pulldown/MS? If not, what was the rationale to focus only on PRSS1? The authors claimed that alanine strongly interacted with PRSS1, however, the binding affinity between them is $2.6 \times 10^{-3} M$. This KD does not denote a strong interaction. The concentration of L-alanine in normal culture medium is only 0.1mM, which means almost no interaction between PRSS1 and alanine under normal cell culture conditions. Accordingly, it should not matter whether Rv2780 degrades alanine or not. More importantly, there is no direct evidence demonstrating that alanine interacts with PRSS1 to induce AMP expression, Fig. 4N is circumstantial, findings can rely on various metabolic pathways. Without evidence of interaction, i.e. mutants, it is impossible to conclude that alanine exerts its

presumed functions by interacting with PRSS1.

3. Potential variability in macrophage cell death upon infection with various Mtb mutants, or upon delivery of various treatments was not at all investigated throughout the manuscript. For instance, the author tested for effects of alanine and GWP-042 on cell viability, but not in the context of Mtb infection. This is a critical confounding factor with major impact on Mtb replication and also on the outcome of the Mtb infection in the mouse model.

Specific comments:

Figure 1: There is no proof for inhibition of the AMP in vivo (transcripts, protein). Rv2780 may have other roles which lead to a better outcome, i.e. lower bacterial loads and tissue inflammation. The CFU values seem uncoupled from tissue inflammation – 10^5 Mtb and almost no lung inflammation is hardly plausible. These findings rather suggest that Rv2780 alters compartmentalization of Mtb in host phagocytes and inflammation.

Figure 2: As indicated above, levels of alanine seem not to differ in infection with various Mtb mutants (statistically not significance). Are levels of alanine in human TB patients correlating with the extent of lung disease (inflammation, chest X-ray grading) or with the smear score (estimation of bacillary load in sputum)? Levels of alanine in the host may be influenced by metabolic pathways irrespective of Mtb load, also in mice. Are Mtb loads at 7 dpi different?

Figure 3: What is the explanation for lower CFU counts in MPM at 24h? Regulation of Defb4 gene occurred only later during infection (48h) (panels C and D). What is the alanine serum level in Mtb-infected WT and KO mice?

Figure 4: As mentioned above, the KD value does not justify binding of L-alanine to PRSS1 in normal culture media. Also in this figure, Mtb CFU values seem uncoupled from tissue inflammation – see comment above.

Figure 5: Supplementation with L-alanine may alter cellular immunometabolism, macrophage viability during Mtb infection and these must be experimentally tested. Delivery of L-alanine to mice during TB infection appears to be one condition which does not result in the uncoupling of the CFU from the inflammation score.

Figure 6: Does GWP-042 activate the CYP450 system? It appears to promote phagocytosis of Mtb by macrophages (panel G). What is the explanation for this observation? What about production of mediators (cytokines, NO) or cell death (as already commented above)?

Supplementary Figures:

1I: The resolution is insufficient; it should be improved.

2G: Why should Mtb signal (GFP) be spread in the cytosol? There are hardly any puncta (mCherry) to see and thus IF quantification is not convincing.

Reviewer #2

Mass spectrometry and metabolomics

(Remarks to the Author):

The study by Peng and colleagues shed lights into a novel mechanism on how *Mycobacterium tuberculosis* can suppress the production of antimicrobial peptides in macrophages via the secretion of an alanine dehydrogenase that then interfere with the alanine levels in macrophages leading to the decrease in expression of those peptides.

The study is very timely and constitute a real novelty in the field by the discovery of Rv2480 as an alanine dehydrogenase that is potentially secreted to impair the pool size of host alanine.

The manuscript is clear and well organised. The experiments have been conducted diligently with appropriate statistical analysis used and controls.

The material and method section is clear and contains the appropriate levels of information.

However, some parts of the study should be clarified:

1) It would be of interest to provide the data for the complement strains for Figure 1C, Figure 1D, Figure 3F to 3G.

2) As rv2480 encodes for an alanine dehydrogenase, it would be of relevance to measure the level of pyruvate in H37Rv infected macrophages and H37Rv Δ rv2480 and complemented strains as well as other metabolites from the central carbon metabolism. That will provide a link between the changes in alanine level and rewiring macrophage metabolome upon infection.

3) To that extend, being able to measure carbon flux through glycolysis and amino acids would be of great interest in terms of antimicrobial peptides metabolism and how Rv2480 interferes with this process.

4) The authors claim that Rv2480 is secreted. It would be good to clearly identify the mechanism on how Rv2480 is secreted and define the stage of the infection that the enzyme is localised, define the compartment, in H37Rv infected macrophages.

5) The fact that L-alanine binds PRSS1 is very interesting, and the authors should further discuss the threshold on the level L-alanine requires to enhance the production of antimicrobial peptide through PRSS1 interaction. Being able to quantify the levels of L-alanine and pyruvate in the different assays employed in this study could be of relevance.

Reviewer #3

Host-pathogen, in vivo animal models, AMPs

(Remarks to the Author):

This study demonstrates for the first time a novel mechanism by which PRSS1 inhibits the NF- κ B pathway, and shows that L-alanine interacts with PRSS1 to unlock PRSS1's inhibition of NF- κ B, thereby promoting AMP expression and release, which in turn kills TB bacteria. Moreover, this effect is achieved by the secretion of Rv2780 by mycobacterium tuberculosis, indicating that bacteria use host metabolism to control host immunity. Finally, A new compound against bacterial Rv2780 was screened and good effect was confirmed.

The work will be of significance to the field. Compared with the existing findings, it is obviously innovative and new, and new drugs are screened for the treatment of tuberculosis and drug-resistant tuberculosis. The work support the conclusions and claims. The methodology is sound. The work meet the expected standards in the field. There are enough supplementary documents provided in the methods for the work to be reproduced.

There are some minor revise suggestions for the MS as follows:

1. Macrophages are heterogeneous, and the functions of tissue-derived macrophages and bone marrow-derived macrophages are also different. Bone marrow-derived macrophages are usually used in studies to produce primary macrophages. In this paper, peritoneal macrophages are used. The reason for using peritoneal macrophages instead of bone marrow-derived macrophages should be explained in the discussion or supplemented in the results.

2. Is the novel mechanism of PRSS1 inhibition of NF- κ B the same in other bacteria? If the same mechanism is involved in the action of bacteria that can produce large amounts of AMP, it is recommended to supplement 1-2 other bacteria

REVIEWER COMMENTS

Reviewer #1

**Expertise - Host pathogen interactions, in vivo murine models
(Remarks to the Author):**

The manuscript by Peng et al. “Pathogenic mycobacteria suppress host antimicrobial peptides by dehydrogenating L-alanine“ reports that a mycobacterial (*M. tuberculosis*, Mtb) secreted protein, notably Rv2780, alters production of AMP in various cells by targeting L-alanine abundance. The authors propose that this amino acid in turn releases the inhibitory effects of the protease serin 1, PRRS1, on NF- κ B-driven AMP production. Supplementation with L-alanine and delivery of a compound which targets Rv2780 showed beneficial effects in the murine TB model. The conclusions are not fully supported by the data.

We are very grateful to the reviewer’s constructive comments and we have addressed these comments as following.

Even the title is misleading because 1) Rv2780 is conserved in non-pathogenic mycobacteria such as *M. smegmatis* (Suppl. Fig. 3), 2) roles beyond AMP have not been tested, and 3) dehydrogenation of L-alanine has not been shown in the infection context. Detailed comments are presented below.

We are deeply sorry for the misleading of the title. 1) We agree that Rv2780 is conserved in non-pathogenic mycobacteria. Thus, the title was revised to “*Mycobacterium tuberculosis* suppresses host antimicrobial peptides by dehydrogenating L-alanine”. 2) As per the reviewer’s suggestion, we examined inflammation cytokines expression (*Il1b*, *Il6*, *Il12* and *Tnf*) in macrophages and lung tissues infected with H37Rv and Δ Rv2780. Although Rv2780 promotes cytokines expression in macrophages (**revised Supplementary Fig. 2I-L**) or lung tissues (**revised Supplementary Fig. 2M-P**) upon *M. tuberculosis* infection, our data showed that deletion of *Defb4* rescued Δ Rv2780 caused decreased *M. tuberculosis* survival in macrophages or lung tissues (**revised Fig. 3H-J**). These data suggested that Rv2780 promotes *M. tuberculosis* survival mainly through inhibiting AMPs, and its promotion of inflammatory cytokines may be a side effect. 3) The dehydrogenation of L-alanine by Rv2780 in the infection context was examined in macrophages and mice lung tissues or serum via alanine assay kit (**revised Fig. 2G-J**). To strength this, we also examine the carbon flux in H37Rv and Δ Rv2780 infected macrophages, showing that Rv2780 may promote hydrolysis of alanine to pyruvate in *Mtb* infected macrophages (**revised Supplementary Fig. 4**).

General comments:

1. Data suggesting a direct inhibition of AMP by Rv2780 are not convincing.

Firstly, alanine is a non-essential amino acid which can be synthesized from pyruvate. The author demonstrated that overexpression of Rv2780 indeed reduced the level of alanine in HEK293T and A549 cells. However, infection of

macrophages with Mtb and Mtb mutants did not result in significant difference of alanine concentration (Fig. 2G, 2H). Therefore, it is questionable whether the amount of Rv2780 during infection can lead to significant reduction of alanine.

We highly appreciate the reviewer's insightful suggestion, which is also mentioned by Reviewer 2#. Rv2780 encodes L-alanine dehydrogenase, an enzyme that catalyzes the NAD⁺-dependent interconversion of alanine and pyruvate. To support this we analyzed total metabolic profiling of macrophages infected with H37Rv, Δ Rv2780, Δ Rv2780+Rv2780 and Δ Rv2780+Rv2780^{DM} for 24 h. Consistent with our previous observation that the level of alanine was reduced in H37Rv or H37Rv (Δ Rv2780+Rv2780) infected macrophages, but infection of H37Rv Δ Rv2780 or H37Rv(Δ Rv2780+Rv2780^{DM}) led to much more abundant alanine in the infected cells (**revised Supplementary Fig. 3L; Supplementary Table 3**). The level of pyruvate was higher in H37Rv or H37Rv (Δ Rv2780+Rv2780) infected macrophages than that of H37Rv Δ Rv2780 or H37Rv (Δ Rv2780+Rv2780^{DM}) infection group (**revised Supplementary Fig. 3L; Supplementary Table 3**). However, deletion Rv2780 had no effects on the content of other central carbon metabolism, including citrate, α -ketoglutarate, fumarate, and lactate (**revised Supplementary Fig. 3M; Supplementary Table 3**). This is consistent with the observation by carbon flux, that Rv2780 had no significant effect on macrophages pyruvate metabolism, except the hydrolysis of alanine to pyruvate (**revised Supplementary Fig. 4; Supplementary Table 4**).

In our previous study, alanine level was normalized to total protein level in each sample to avoid the influence of different cell numbers. However, during the process of pathogen infection the host protein synthesis function of ribosomes are altered (**Jiao et al., 2023; Garofalo et al., 2023**). Thus it would be more accurate to normalize the intracellular alanine level to GAPDH, an indicator that can reflect the number of cells. Actually, the exact alanine concentration was significantly increased in Δ Rv2780-infected macrophage by 2.27-fold in lysates and 2.50-fold in supernatants comparing with that of H37Rv at 24 hours post infection (**revised Fig. 2G, H**), which suggested that Rv2780 during infection can lead to significant reduction of alanine. This was also supported by LC-MS analysis showing 3.69-fold increase of L-alanine by Δ Rv2780 infection compared with that of H37Rv at 24 hours post infection (**revised Supplementary Fig. 3L; Supplementary Table 3**).

Levels of Rv2780 at 3 and 6 hpi in MPM lysates are provided and seem impressive (Suppl. Fig. 1). yet it remains questionable how Mtb can produce high amounts of likely secreted protein in such a short time.

We appreciate the reviewer's questions. We detected abundant Rv2780 protein in *M. tuberculosis* H37Rv infected macrophages from 3 to 6 hours after infection, which is consistent with other *M. tuberculosis* secreted proteins, like PtpA (**Wang et al 2015**), PknG (**Ge et al., 2022**), UreC (**Liu et al., 2023**), Rv0222(**Wang et al., 2020**), Mpt53(**Wang et al 2019**), et al.

To adapt to host microenvironment, like hypoxia, or reactive nitrogen species or

low pH, mycobacteria enhance widespread genes transcription and secretion (Yang et al., 2021; Galagan et al., 2013; Bi et al., 2018; Yang et al., 2018; Birhanu et al., 2022; Fisher et al., 2002; Rohde et al., 2007). Notable *M. tuberculosis* transcriptional induction was detected within minutes post infection in phagosome of macrophages (Rohde et al., 2007). Furthermore, the expression of the mycobacterial *ald* gene (Rv2780) is strongly upregulated by nutrient starvation, hypoxia or in granuloma (Betts et al., 2002; Giffin et al., 2016; Kaman et al., 2002; Jeong et al., 2015). It is possible that in response to host macrophages microenvironment, the expression of Rv2780 was induced.

2. The interaction between PRSS1 and alanine is arguable.

Is PRSS1 the only hit in the pulldown/MS? If not, what was the rationale to focus only on PRSS1?

We appreciate the reviewer's questions. PRSS1 is not the only hit in the pulldown/MS, and the other proteins were shown in **Supplementary Table 5**. However, from all the identified binding proteins, we found only *Prss1* deficiency could rescue L-alanine mediated inhibition of intracellular survival of *M. tuberculosis* in macrophages (**revised Fig. 4N, O**), suggesting that L-alanine may restrict the intracellular growth of *M. tuberculosis* through PRSS1. Thus we focus only on PRSS1 for further exploration.

The authors claimed that alanine strongly interacted with PRSS1, however, the binding affinity between them is $2.6 \times 10^{-3} \text{M}$. This KD does not denote a strong interaction. The concentration of L-alanine in normal culture medium is only 0.1mM, which means almost no interaction between PRSS1 and alanine under normal cell culture conditions. Accordingly, it should not matter whether Rv2780 degrades alanine or not.

We highly appreciate the reviewer's insightful comments. Surface plasmon resonance (SPR) assay revealed that the binding affinity between PRSS1 and L-alanine is $2.6 \times 10^{-3} \text{M}$ (**previous Fig. 4D**). During SPR assay, PRSS1 protein was fixed on the COOH sensor chip by capture-coupling, then L-alanine was injected sequentially into the chamber in PBS at 25°C. During this process, PRSS1 protein fixation and the reaction buffer may affect the binding result. Thus we applied MicroScale Thermophoresis (MST) analysis, which has no fixed requirement and buffer limitation (**Wienken et al., 2010**), to reanalyze the interaction of PRSS1 protein and L-alanine. Via MST, we found the binding affinity between PRSS1 and L-alanine is $8.88 \times 10^{-5} \text{M}$ in assay buffer (50mM HEPES, pH7.5, 500mM NaCl, 5% Glycerol, 1mM TCEP) (**revised Fig. 4**), which is much lower than the concentration of L-alanine in normal culture medium.

More importantly, there is no direct evidence demonstrating that alanine interacts with PRSS1 to induce AMP expression, Fig. 4N is circumstantial, findings can rely on various metabolic pathways. Without evidence of interaction, i.e. mutants, it is impossible to conclude that alanine exerts its presumed functions by interacting with PRSS1.

We highly appreciate the reviewer's critical comments. Through *Prss1*^{+/-} peritoneal

macrophages, we found L-alanine promotes AMPs expression and inhibits the intracellular survival of *M. tuberculosis* H37Rv through PRSS1 (**revised Fig. 4J, 4N, 4O; revised Supplementary Fig. 7O-Q**). We did not identify the PRSS1 mutant that does not interact with L-alanine, therefore it is not accurate to make the conclusion that “L-Alanine interacts with PRSS1 to induce AMPs”. It is more accurate to divide this conclusion into two parts, as “L-Alanine interacts with PRSS1” and “L-alanine induces AMPs via PRSS1” (Line 276 and Line 320).

3. Potential variability in macrophage cell death upon infection with various Mtb mutants, or upon delivery of various treatments was not at all investigated throughout the manuscript. For instance, the author tested for effects of alanine and GWP-042 on cell viability, but not in the context of Mtb infection. This is a critical confounding factor with major impact on Mtb replication and also on the outcome of the Mtb infection in the mouse model.

Thanks to the reviewer’s concern. We tested cell viability in macrophages infected with various *Mtb* mutants including H37Rv, Δ Rv2780, Δ Rv2780+Rv2780 and Δ Rv2780+Rv2780^{DM}. As shown in **revised Supplementary Fig. 6A**, various Rv2780 mutants of *Mtb* have little impacts on cell viability during *Mtb* infection. To evaluate the effect of L-alanine and GWP-042 on cell viability in the context of *Mtb* infection, we also tested cell viability in H37Rv-infected macrophages pretreated with various concentration of L-alanine or GWP-042. It shows that L-alanine and GWP-042 don’t affect cell viability during *Mtb* infection (**revised Supplementary Fig. 8D; revised Supplementary Fig. 10K**).

Specific comments:

Figure 1: There is no proof for inhibition of the AMP in vivo (transcripts, protein). Rv2780 may have other roles which lead to a better outcome, i.e. lower bacterial loads and tissue inflammation.

Thanks to the reviewer’s comments. To examine the inhibition of Rv2780 on AMPs *in vivo*, we detected Defb4 and Camp protein level in the serum and lung tissues of mice infected with H37Rv, Δ Rv2780, Δ Rv2780+Rv2780 and Δ Rv2780+Rv2780^{DM}. As shown in **revised Fig. 3D-E** and **revised Supplementary Fig. 5K-L**, Rv2780 also inhibited AMPs *in vivo* and in an alanine dehydrogenase activity dependent manner.

The CFU values seem uncoupled from tissue inflammation – 10^5 Mtb and almost no lung inflammation is hardly plausible. These findings rather suggest that Rv2780 alters compartmentalization of Mtb in host phagocytes and inflammation. We appreciate the reviewer’s concern. As shown in **previous Fig. 1 F-G**, the inflammatory infiltration area of lung tissues from mice infected with H37Rv Δ Rv2780 exhibited ~10% inflammatory infiltration area with 10^5 Mtb CFU suggesting a certain degree of inflammation. This is consistent with recent study, showing 10^5 Mtb CFU values coupled from ~10% inflammatory infiltration area in lung tissues (**Chai et al., 2022**).

As mentioned above, we examined inflammation cytokines expression (*Il1 β* , *Il6*, *Il12*

and *Tnfα*) in lung tissues infected with H37Rv and ΔRv2780. Although Rv2780 promotes cytokines expression in macrophages or lung tissues upon *M. tuberculosis* infection (**revised Supplementary Fig. 2I-P**), our previous data showed that deletion of *Defb4* rescued ΔRv2780 caused decrease of *M. tuberculosis* survival in macrophages (**Fig. 3H, 3I**) and mice lung tissues (**revised Fig. 3J-L**). These data suggested that Rv2780 promotes *M. tuberculosis* survival mainly through inhibiting AMPs, and its promotion of inflammatory cytokines may be a side effect.

Figure 2: As indicated above, levels of alanine seem not to differ in infection with various Mtb mutants (statistically not significance).

We appreciate the reviewer's concern. As mentioned above, we have normalized the intracellular alanine level to cell viability. Actually, the exact alanine concentration was significantly increased in ΔRv2780-infected macrophage by 2.27-fold in lysates and 2.50-fold in supernatants comparing with that of H37Rv at 24 hours post infection (**revised Fig. 2G, H**), which suggested that Rv2780 during infection can lead to significant reduction of alanine. This was also supported by LC-MS analysis showing 3.69-fold increase of L-alanine by ΔRv2780 infection compared with that of H37Rv at 24 hours post infection (**revised Supplementary Fig. 3L; Supplementary Table 3**).

Are levels of alanine in human TB patients correlating with the extent of lung disease (inflammation, chest X-ray grading) or with the smear score (estimation of bacillary load in sputum)?

We appreciate the reviewer's question. Among the 35 TB patients collected from our previous study (**previous Supplementary Fig. 3D**), we compared the chest X-ray score and the smear score between the top 7 patients with the highest plasma alanine level and the bottom 7 patients. We found TB patients with lower alanine level exhibited a trend of more severe pulmonary pathological damage, indicated by higher X-ray score (**revised Supplementary Fig. 3F, 3G**). However, it seems that alanine level is not correlated with the smear score (**revised Supplementary Fig. 3H**). This may be because the smear score cannot fully reflect the bacterial load in TB patients.

Levels of alanine in the host may be influence by metabolic pathways irrespective of Mtb load, also in mice. Are Mtb loads at 7 dpi different?

We appreciate the reviewer's question. *Mtb* loads is lower in lung tissues from ΔRv2780-infected mice at 7 dpi (**revised Fig. 3J**).

Figure 3: What is the explanation for lower CFU counts in MPM at 24h? Regulation of Defb4 gene occurred only later during infection (48h) (panels C and D).

We appreciate the reviewer's comments. In our previous study, we only examined the protein level of *Defb4* intracellularly. But antimicrobial peptides are reported to be secreted out of the cell (**Izadpanah et al., 2004; Yeaman et al., 2003**). To access the total *Defb4* in H37Rv, ΔRv2780, ΔRv2780+Rv2780 and ΔRv2780+Rv2780^{DM}-infected macrophages, we also examined *Defb4* protein level in supernatants and found

that total protein level of Defb4 is slightly increased in cell lysates but significantly increased in supernatants of Δ Rv2780-infected macrophages at 24 hours post infection (**revised Fig. 3C** and **revised Supplementary Fig. 5G**), thus the regulation of Defb4 gene by Rv2780 occurred at 24 hours post infection.

What is the alanine serum level in Mtb-infected WT and KO mice?

We appreciate the reviewer's question. No difference of the alanine level in serum and lung tissues was detected between WT and *Defb4*^{-/-} mice when infected with H37Rv (**revised Supplementary Fig. 6H, 6I**).

Figure 4: As mentioned above, the KD value does not justify binding of L-alanine to PRSS1 in normal culture media.

Thanks to the reviewer's insightful comments. As mentioned above we examined the interaction of PRSS1 protein and L-alanine via MicroScale Thermophoresis (MST) analysis (**Wienken et al., 2010**). Using MST analysis, we found the binding affinity between PRSS1 and L-alanine is 8.88×10^{-5} M in assay buffer (50mM HEPES, pH7.5, 500mM NaCl, 5% Glycerol, 1mM TCEP) (**revised Fig. 4D**), which is much lower than the concentration of L-alanine in normal culture medium.

Also in this figure, Mtb CFU values seem uncoupled from tissue inflammation – see comment above.

We appreciate the reviewer's comments. 10^5 Mtb CFU values coupled from ~10% inflammatory infiltration area in lung tissues (**Chai et al., 2022**). Thus we replaced the data of haematoxylin & eosin staining and histological score to another more representative results in the revised figure (**revised Fig. 3K, 3L**).

Figure 5: Supplementation with L-alanine may alter cellular immunometabolism, macrophage viability during Mtb infection and these must be experimentally tested.

Thanks to the reviewer's concern. We have tested macrophage cell viability in H37Rv-infected macrophages pretreated with various concentration of L-alanine, and found supplementation with L-alanine has no effect on macrophage viability during Mtb infection (**revised Supplementary Fig. 8D**).

Delivery of L-alanine to mice during TB infection appears to be one condition which does not result in the uncoupling of the CFU from the inflammation score.

We appreciate the reviewer's comments. In **revised Fig. 5F-H**, 10^4 Mtb CFU values coupled from ~50% inflammatory infiltration area in lung tissues from mice supplemented with L-alanine, which is different with other mice infection experiments in this study (**revised Fig. 1** and **revised Fig. 3**). We delivered L-alanine to mice by drinking water, which is absorbed through the digestive tract and may affect gut microbiota. Considered gut microbiota plays a critical role in regulating host anti-TB immunity (**Yang et al., 2022; Wipperman et al., 2021**), it is possible that delivered L-alanine to mice by drinking water may have some additional effects on lung CFU or the

lung inflammation upon Mtb infection.

Figure 6: Does GWP-042 activate the CYP450 system?

We appreciate the reviewer's question. Referring to the method used in recently study (Wang et al., 2023), we examined the expression of cytochrome P450 (CYP450) homologs, *Cyp1a2*, *Cyp2b10*, *Cyp2c38*, *Cyp2d9*, and *Cyp3a11*, in murine hepatocytes. Meanwhile, rifampicin was used as positive control, which is frequently used as a positive control or calibrator for evaluating the CYP450 activation of tested compounds (Kenny et al., 2018). As shown in revised Supplementary Fig. 10M, rifampicin significantly induced these genes expression, while GWP-042 could not. These data suggesting GWP-042 may not activate the CYP450 system.

It appears to promote phagocytosis of Mtb by macrophages (panel G). What is the explanation for this observation?

We appreciate the reviewer's question. In revised Fig. 6G, H37Rv and H37RvΔRv2780 exhibited similar intracellular CFU at 3h post infection, suggesting Rv2780 alone has no effect on macrophages phagocytosis. Higher H37Rv CFU was detected in macrophages treated with GWP-042 at 3h post infection, notably this was not observed in H37RvΔRv2780 infection. One possibility is that GWP-042-Rv2780 protein complex, which only existed in GWP-042 treated H37Rv infection condition, may additionally promote macrophages phagocytosis.

What about production of mediators (cytokines, NO) or cell death (as already commented above)?

We appreciate the reviewer's question. GWP-042 has no significant effect on macrophages cytokines expression, NO production or cell death (revised Supplementary Fig. 10A, 10B, 10K).

Supplementary Figures:

1I: The resolution is insufficient; it should be improved.

Thanks to the reviewer's comments. We have enlarged and replaced the images in revised manuscript (revised Supplementary Fig. 1I).

2G: Why should Mtb signal (GFP) be spread in the cytosol? There are hardly any puncta (mCherry) to see and thus IF quantification is not convincing.

Thanks to the reviewer's comments. Revised Supplementary Figure 2G shows the macrophage transfected with adenovirus expressing mCherry-GFP-LC3B fusion protein to visualize free autophagosomes (GFP and mCherry fluorescence). Only mCherry fluorescence retained when the autophagosomes have fused with the lysosome, due to acid sensitivity of GFP (Li et al., 2022; Geng et al., 2020). Green fluorescence shown in the figure is GFP protein not quenched by acidic lysosome. To clearly show the puncta in the cytosol, we have enlarged and replaced the images in revised figure (revised Supplementary Fig. 2G)

Reviewer #2

Mass spectrometry and metabolomics

(Remarks to the Author):

The study by Peng and colleagues shed lights into a novel mechanism on how Mycobacterium tuberculosis can suppress the production of antimicrobial peptides in macrophages via the secretion of an alanine dehydrogenase that then interfere with the alanine levels in macrophages leading to the decrease in expression of those peptides.

The study is very timely and constitute a real novelty in the field by the discovery of Rv2480 as an alanine dehydrogenase that is potentially secreted to impair the pool size of host alanine.

We highly appreciate the reviewer for the positive comments and the affirmation of the novelty of our work.

The manuscript is clear and well organized. The experiments have been conducted diligently with appropriate statistical analysis used and controls.

We thank the reviewer for recognizing our research work.

The material and method section is clear and contains the appropriate levels of information.

We thank the reviewer for recognizing our research work.

However, some parts of the study should be clarified:

We are very grateful to the reviewer's constructive comments and suggestions. We have addressed these comments as following.

1) It would be of interest to provide the data for the complement strains for Figure 1C, Figure 1D, Figure 3F to 3G.

Thanks to the reviewer's suggestion. We have provided the data for the complement strains for previous Figure 1C, 1D, Figure 3F to 3G in the revised manuscript (**revised Fig. 1C, 1D and revised Fig. 3F-G**).

2) As rv2480 encodes for an alanine dehydrogenase, it would be of relevance to measure the level of pyruvate in H37Rv infected macrophages and H37Rv Δ rv2480 and complemented strains as well as other metabolites from the central carbon metabolism. That will provide a link between the changes in alanine level and rewiring macrophage metabolome upon infection.

We highly appreciate the reviewer's insightful suggestion, which is also mentioned by Reviewer 1#. Rv2780 encodes L-alanine dehydrogenase, an enzyme that catalyzes the NAD⁺-dependent interconversion of alanine and pyruvate. To support this we analyzed total metabolic profiling of macrophages infected with H37Rv, Δ Rv2780, Δ Rv2780+Rv2780 and Δ Rv2780+Rv2780^{DM} for 24 h. Consistent with our previous observation that the level of alanine was reduced in H37Rv or H37Rv

(Δ Rv2780+Rv2780) infected macrophages, but infection of H37Rv Δ Rv2780 or H37Rv(Δ Rv2780+Rv2780^{DM}) led to much more abundant alanine in the infected cells (**revised Supplementary Fig. 3L; Supplementary Table 3**). The level of pyruvate was higher in H37Rv or H37Rv (Δ Rv2780+Rv2780) infected macrophages than that of H37Rv Δ Rv2780 or H37Rv (Δ Rv2780+Rv2780^{DM}) infection group (**revised Supplementary Fig. 3L; Supplementary Table 3**). However, deletion Rv2780 had no effects on the content of other central carbon metabolism, including citrate, α -ketoglutarate, fumarate, and lactate (**revised Supplementary Fig. 3M; Supplementary Table 3**). This is consistent with the observation by carbon flux, that Rv2780 had no significant effect on macrophages pyruvate metabolism, except the hydrolysis of alanine to pyruvate (**revised Supplementary Fig. 4; Supplementary Table 4**).

3) To that extend, being able to measure carbon flux through glycolysis and amino acids would be of great interest in terms of antimicrobial peptides metabolism and how Rv2480 interferes with this process.

Thanks to the reviewer's suggestion. We measured carbon flux through glycolysis and amino acids in H37Rv and Δ Rv2780-infected macrophages to determine the process of carbon metabolism during *Mtb* infection and how Rv2780 interferes with this process. ¹³C6-glucose is converted to produce labeled pyruvate, which can be decarboxylated to form labeled two-carbon metabolite acetyl-CoA and entered into TCA cycle, while unlabeled alanine is dehydrogenized by Rv2780 to form unlabeled pyruvate. Pyruvate provides two carbons to acetyl-CoA, citrate, α -ketoglutarate, succinate and fumarate. Both labeled and unlabeled pyruvate-derived acetyl-CoA are entered into the TCA cycle respectively. Compared with Δ Rv2780 infection, the percentage of unlabeled alanine is decreased in H37Rv-infected macrophage suggesting the metabolic flux from alanine to pyruvate in the presence of Rv2780 (**revised Supplementary Fig. 4**). However, the percentage of unlabeled of other metabolites were not significantly different between H37Rv- and Δ Rv2780-infected macrophages. These results suggested that Rv2780 has no significant effect on macrophages glycolysis or TCA cycle.

4) The authors claim that Rv2480 is secreted. It would be good to clearly identify the mechanism on how Rv2480 is secreted and define the stage of the infection that the enzyme is localised, define the compartment, in H37Rv infected macrophages.

Thanks to the reviewer's suggestion. To analyze subcellular localization of Rv2780 during *Mtb* infection, we detected Rv2780 by immunofluorescence microscopy. Rv2780 was mainly detected in the cytoplasm, partially in mitochondria, very minimally in the endoplasmic reticulum (ER) or lysosome (**revised Supplementary Fig. 1I**).

5) The fact that L-alanine binds PRSS1 is very interesting, and the authors should further discuss the threshold on the level L-alanine requires to enhance the production of antimicrobial peptide through PRSS1 interaction. Being able to quantify the levels of L-alanine and pyruvate in the different assays employed in

this study could be of relevance.

We appreciate the reviewer's comments. The binding affinity between PRSS1 and L-alanine is $887.8 \pm 27.1 \mu\text{M}$ by MicroScale Thermophoresis analysis, suggesting a strong interaction. To determine the threshold of L-alanine level to enhance antimicrobial peptide through PRSS1, we supplemented WT and *Prss1*^{+/-} macrophage with different concentrations of L-alanine, and found 0.01mM L-alanine is sufficient to induce *Defb4* expression in WT, but not *Prss1*^{+/-} MPMs (**revised Supplementary Fig. 7Q**).

Reviewer #3

Host-pathogen, in vivo animal models, AMPs

(Remarks to the Author):

This study demonstrates for the first time a novel mechanism by which PRSS1 inhibits the NF- κ B pathway, and shows that L-alanine interacts with PRSS1 to unlock PRSS1's inhibition of NF- κ B, thereby promoting AMP expression and release, which in turn kills TB bacteria. Moreover, this effect is achieved by the secretion of Rv2780 by mycobacterium tuberculosis, indicating that bacteria use host metabolism to control host immunity. Finally, A new compound against bacterial Rv2780 was screened and good effect was confirmed.

We highly appreciate the reviewer for the positive comments and the affirmation of the novelty of our work.

The work will be of significance to the field. Compared with the existing findings, it is obviously innovative and new, and new drugs are screened for the treatment of tuberculosis and drug-resistant tuberculosis. The work support the conclusions and claims. The methodology is sound. The work meet the expected standards in the field. There are enough supplementary documents provided in the methods for the work to be reproduced.

We thank the reviewer for recognizing our research work.

There are some minor revise suggestions for the MS as follows:

We are very grateful to the reviewer's constructive comments and suggestions. We have addressed these comments as following.

1. Macrophages are heterogeneous, and the functions of tissue-derived macrophages and bone marrow-derived macrophages are also different. Bone marrow-derived macrophages are usually used in studies to produce primary macrophages. In this paper, peritoneal macrophages are used. The reason for using peritoneal macrophages instead of bone marrow-derived macrophages should be explained in the discussion or supplemented in the results.

We appreciate the reviewer's comments. Peritoneal macrophages are also widely used to study the functional role and mechanism of pathogenic bacteria secretory proteins on host immune response, indicating the existence of integral immune molecules in peritoneal macrophages (Zhang et al., 2011; Wang et al., 2015; Wang et al., 2019). To further examine whether our discovery is universal in different macrophages, we also examined AMPs expression and *Mtb* intracellular survival in H37Rv, Δ Rv2780, Δ Rv2780+Rv2780 and Δ Rv2780+Rv2780^{DM} infected bone marrow-derived macrophages (BMDMs). Consistent with peritoneal macrophages, Rv2780 also inhibited AMPs expression and promoted *Mtb* intracellular survival through its alanine dehydrogenase activity in BMDMs (revised Supplementary Fig. 5I, 5J; revised Supplementary Fig. 6B, 6C).

2. Is the novel mechanism of PRSS1 inhibition of NF- κ B the same in other bacteria? If the same mechanism is involved in the action of bacteria that can produce large amounts of AMP, it is recommended to supplement 1-2 other bacteria.

Thanks to the reviewer's insightful question. We observed much higher level of NF- κ B activation and AMPs expression in *Prss1*^{+/-} macrophages in response to gram-negative bacteria *E. coli* (**revised Supplementary Fig. 7G, 7I-K**) or another gram-positive bacteria *S. aureus* infection (**revised Supplementary Fig. 7H, 7L-N**). These data suggesting the PRSS1 inhibition of NF- κ B mediated AMPs expression may be a general mechanism.

References:

- Betts JC, Lukey PT, Robb LC, McAdam RA, Duncan K. Evaluation of a nutrient starvation model of Mycobacterium tuberculosis persistence by gene and protein expression profiling. *Mol Microbiol.* 2002 Feb;43(3):717-31. doi: 10.1046/j.1365-2958.2002.02779.x. PMID: 11929527.
- Bi J, Gou Z, Zhou F, Chen Y, Gan J, Liu J, Wang H, Zhang X. Acetylation of lysine 182 inhibits the ability of Mycobacterium tuberculosis DosR to bind DNA and regulate gene expression during hypoxia. *Emerg Microbes Infect.* 2018 Jun 13;7(1):108. doi: 10.1038/s41426-018-0112-3. PMID: 29899473; PMCID: PMC5999986.
- Birhanu AG, Gómez-Muñoz M, Kalayou S, Riaz T, Lutter T, Yimer SA, Abebe M, Tønjum T. Proteome Profiling of Mycobacterium tuberculosis Cells Exposed to Nitrosative Stress. *ACS Omega.* 2022 Jan 14;7(4):3470-3482. doi: 10.1021/acsomega.1c05923. PMID: 35128256; PMCID: PMC8811941.
- Chai Q, Yu S, Zhong Y, Lu Z, Qiu C, Yu Y, Zhang X, Zhang Y, Lei Z, Qiang L, Li BX, Pang Y, Qiu XB, Wang J, Liu CH. A bacterial phospholipid phosphatase inhibits host pyroptosis by hijacking ubiquitin. *Science.* 2022 Oct 14;378(6616):eabq0132. doi: 10.1126/science.abq0132. Epub 2022 Oct 14. PMID: 36227980.
- Galagan JE, Minch K, Peterson M, Lyubetskaya A, Azizi E, Sweet L, Gomes A, Rustad T, Dolganov G, Glotova I, Abeel T, Mahwinney C, Kennedy AD, Allard R, Brabant W, Krueger A, Jaini S, Honda B, Yu WH, Hickey MJ, Zucker J, Garay C, Weiner B, Sisk P, Stolte C, Winkler JK, Van de Peer Y, Iazzetti P, Camacho D, Dreyfuss J, Liu Y, Dorhoi A, Mollenkopf HJ, Drogaris P, Lamontagne J, Zhou Y, Piquenot J, Park ST, Raman S, Kaufmann SH, Mohny RP, Chelsky D, Moody DB, Sherman DR, Schoolnik GK. The Mycobacterium tuberculosis regulatory network and hypoxia. *Nature.* 2013 Jul 11;499(7457):178-83. doi: 10.1038/nature12337. Epub 2013 Jul 3. PMID: 23823726; PMCID: PMC4087036.
- Garofalo M, Payros D, Taieb F, Oswald E, Nougayrède JP, Oswald IP. From ribosome to ribotoxins: understanding the toxicity of deoxynivalenol and Shiga toxin, two food borne toxins. *Crit Rev Food Sci Nutr.* 2023 Oct 20:1-13. doi:

- 10.1080/10408398.2023.2271101. Epub ahead of print. PMID: 37862145.
- Geng N, Wang X, Yu X, Wang R, Zhu Y, Zhang M, Liu J, Liu Y. Staphylococcus aureus Avoids Autophagy Clearance of Bovine Mammary Epithelial Cells by Impairing Lysosomal Function. *Front Immunol.* 2020 May 5;11:746. doi: 10.3389/fimmu.2020.00746. PMID: 32431700; PMCID: PMC7214833.
- Ge P, Lei Z, Yu Y, Lu Z, Qiang L, Chai Q, Zhang Y, Zhao D, Li B, Pang Y, Liu CH, Wang J. M. tuberculosis PknG manipulates host autophagy flux to promote pathogen intracellular survival. *Autophagy.* 2022 Mar;18(3):576-594. doi: 10.1080/15548627.2021.1938912. Epub 2021 Jul 7. PMID: 34092182; PMCID: PMC9037497.
- Giffin MM, Shi L, Gennaro ML, Sohaskey CD. Role of Alanine Dehydrogenase of Mycobacterium tuberculosis during Recovery from Hypoxic Nonreplicating Persistence. *PLoS One.* 2016 May 20;11(5):e0155522. doi: 10.1371/journal.pone.0155522. PMID: 27203084; PMCID: PMC4874671.
- Li L, Cui YJ, Liu Y, Li HX, Su YD, Li SN, Wang LL, Zhao YW, Wang SX, Yan F, Dong B. ATP6AP2 knockdown in cardiomyocyte deteriorates heart function via compromising autophagic flux and NLRP3 inflammasome activation. *Cell Death Discov.* 2022 Apr 4;8(1):161. doi: 10.1038/s41420-022-00967-w. PMID: 35379787; PMCID: PMC8980069.
- Liu S, Guan L, Peng C, Cheng Y, Cheng H, Wang F, Ma M, Zheng R, Ji Z, Cui P, Ren Y, Li L, Shi C, Wang J, Huang X, Cai X, Qu D, Zhang H, Mao Z, Liu H, Wang P, Sha W, Yang H, Wang L, Ge B. Mycobacterium tuberculosis suppresses host DNA repair to boost its intracellular survival. *Cell Host Microbe.* 2023 Nov 8;31(11):1820-1836.e10. doi: 10.1016/j.chom.2023.09.010. Epub 2023 Oct 16. PMID: 37848028.
- Jeong JA, Hyun J, Oh JI. Regulation Mechanism of the ald Gene Encoding Alanine Dehydrogenase in Mycobacterium smegmatis and Mycobacterium tuberculosis by the Lrp/AsnC Family Regulator AldR. *J Bacteriol.* 2015 Oct;197(19):3142-53. doi: 10.1128/JB.00453-15. Epub 2015 Jul 20. PMID: 26195594; PMCID: PMC4560286.
- Jiao L, Liu Y, Yu XY, Pan X, Zhang Y, Tu J, Song YH, Li Y. Ribosome biogenesis in disease: new players and therapeutic targets. *Signal Transduct Target Ther.* 2023 Jan 9;8(1):15. doi: 10.1038/s41392-022-01285-4. PMID: 36617563; PMCID: PMC9826790.
- Chan K, Knaak T, Satkamp L, Humbert O, Falkow S, Ramakrishnan L. Complex pattern of Mycobacterium marinum gene expression during long-term granulomatous infection. *Proc Natl Acad Sci U S A.* 2002 Mar 19;99(6):3920-5. doi: 10.1073/pnas.002024599. Epub 2002 Mar 12. PMID: 11891270; PMCID: PMC122624.

- Kenny JR, Ramsden D, Buckley DB, Dallas S, Fung C, Mohutsky M, Einolf HJ, Chen L, Dekeyser JG, Fitzgerald M, Goosen TC, Siu YA, Walsky RL, Zhang G, Tweedie D, Hariparsad N. Considerations from the Innovation and Quality Induction Working Group in Response to Drug-Drug Interaction Guidances from Regulatory Agencies: Focus on CYP3A4 mRNA In Vitro Response Thresholds, Variability, and Clinical Relevance. *Drug Metab Dispos.* 2018 Sep;46(9):1285-1303. doi: 10.1124/dmd.118.081927. Epub 2018 Jun 29. PMID: 29959133.
- Rohde KH, Abramovitch RB, Russell DG. Mycobacterium tuberculosis invasion of macrophages: linking bacterial gene expression to environmental cues. *Cell Host Microbe.* 2007 Nov 15;2(5):352-64. doi: 10.1016/j.chom.2007.09.006. PMID: 18005756.
- Wang G, Li J, Bojmar L, Chen H, Li Z, Tobias GC, Hu M, Homan EA, Lucotti S, Zhao F, Posada V, Oxley PR, Cioffi M, Kim HS, Wang H, Lauritzen P, Boudreau N, Shi Z, Burd CE, Zippin JH, Lo JC, Pitt GS, Hernandez J, Zambirinis CP, Hollingsworth MA, Grandgenett PM, Jain M, Batra SK, DiMaio DJ, Grem JL, Klute KA, Trippett TM, Egeblad M, Paul D, Bromberg J, Kelsen D, Rajasekhar VK, Healey JH, Matei IR, Jarnagin WR, Schwartz RE, Zhang H, Lyden D. Tumour extracellular vesicles and particles induce liver metabolic dysfunction. *Nature.* 2023 Jun;618(7964):374-382. doi: 10.1038/s41586-023-06114-4. Epub 2023 May 24. PMID: 37225988; PMCID: PMC10330936.
- Wang J, Li BX, Ge PP, Li J, Wang Q, Gao GF, Qiu XB, Liu CH. Mycobacterium tuberculosis suppresses innate immunity by coopting the host ubiquitin system. *Nat Immunol.* 2015 Mar;16(3):237-45. doi: 10.1038/ni.3096. Epub 2015 Feb 2. PMID: 25642820.
- Wang L, Liu Z, Wang J, Liu H, Wu J, Tang T, Li H, Yang H, Qin L, Ma D, Chen J, Liu F, Wang P, Zheng R, Song P, Zhou Y, Cui Z, Wu X, Huang X, Liang H, Zhang S, Cao J, Wu C, Chen Y, Su D, Chen X, Zeng G, Ge B. Oxidization of TGF β -activated kinase by MPT53 is required for immunity to Mycobacterium tuberculosis. *Nat Microbiol.* 2019 Aug;4(8):1378-1388. doi: 10.1038/s41564-019-0436-3. Epub 2019 May 20. PMID: 31110366.
- Wang L, Wu J, Li J, Yang H, Tang T, Liang H, Zuo M, Wang J, Liu H, Liu F, Chen J, Liu Z, Wang Y, Peng C, Wu X, Zheng R, Huang X, Ran Y, Rao Z, Ge B. Host-mediated ubiquitination of a mycobacterial protein suppresses immunity. *Nature.* 2020 Jan;577(7792):682-688. doi: 10.1038/s41586-019-1915-7. Epub 2020 Jan 15. PMID: 31942069.
- Wipperman MF, Bhattarai SK, Vorkas CK, Maringati VS, Taur Y, Mathurin L, McAulay K, Vilbrun SC, Francois D, Bean J, Walsh KF, Nathan C, Fitzgerald DW, Glickman MS, Bucci V. Gastrointestinal microbiota composition predicts peripheral inflammatory state during treatment of human tuberculosis. *Nat Commun.* 2021

Feb 18;12(1):1141. doi: 10.1038/s41467-021-21475-y. PMID: 33602926; PMCID: PMC7892575.

- Yang F, Yang Y, Chen L, Zhang Z, Liu L, Zhang C, Mai Q, Chen Y, Chen Z, Lin T, Chen L, Guo H, Zhou L, Shen H, Chen X, Liu L, Zhang G, Liao H, Zeng L, Zeng G. The gut microbiota mediates protective immunity against tuberculosis via modulation of lncRNA. *Gut Microbes*. 2022 Jan-Dec;14(1):2029997. doi: 10.1080/19490976.2022.2029997. PMID: 35343370; PMCID: PMC8966992.
- Wienken CJ, Baaske P, Rothbauer U, Braun D, Duhr S. Protein-binding assays in biological liquids using microscale thermophoresis. *Nat Commun*. 2010 Oct 19;1:100. doi: 10.1038/ncomms1093. PMID: 20981028.
- Yang H, Wang F, Guo X, Liu F, Liu Z, Wu X, Zhao M, Ma M, Liu H, Qin L, Wang L, Tang T, Sha W, Wang Y, Chen J, Huang X, Wang J, Peng C, Zheng R, Tang F, Zhang L, Wu C, Oehlers SH, Song Z, She J, Feng H, Xie X, Ge B. Interception of host fatty acid metabolism by mycobacteria under hypoxia to suppress anti-TB immunity. *Cell Discov*. 2021 Oct 5;7(1):90. doi: 10.1038/s41421-021-00301-1. PMID: 34608123; PMCID: PMC8490369.
- Yang H, Sha W, Liu Z, Tang T, Liu H, Qin L, Cui Z, Chen J, Liu F, Zheng R, Huang X, Wang J, Feng Y, Ge B. Lysine acetylation of DosR regulates the hypoxia response of *Mycobacterium tuberculosis*. *Emerg Microbes Infect*. 2018 Mar 21;7(1):34. doi: 10.1038/s41426-018-0032-2. PMID: 29559631; PMCID: PMC5861037.
- Zhang L, Ding X, Cui J, Xu H, Chen J, Gong YN, Hu L, Zhou Y, Ge J, Lu Q, Liu L, Chen S, Shao F. Cysteine methylation disrupts ubiquitin-chain sensing in NF- κ B activation. *Nature*. 2011 Dec 11;481(7380):204-8. doi: 10.1038/nature10690. PMID: 22158122.

REVIEWER COMMENTS

Reviewer #1 (Remarks to the Author):

The authors invested substantial efforts to improve the manuscript.

Statistics and methodology related to normalization of alanine concentrations in macrophages should be provided in the manuscript (materials and methods, legend Fig. 2 G,H).

The Prss1 KO phenotype is likely strongly NF-kB-related and not only AMP-related. Accordingly, the observed very strong phenotype in TB (and other infections) (Fig. 4). These observations should be at least acknowledged and accordingly discussed.

The titles of the figures should be carefully checked for accuracy. Fig. 1 does not show that Rv2780 inhibits AMPs, rather that the mutant is attenuated in macrophages and mice. The title for Fig. 4 is a statement that the authors themselves indicated as non-accurate "L-alanine interacts with PRSS1 to induce Defb4 expression" (similar to Fig. S7 L-alanine interacts with PRSS1 to induce AMPs) (see authors' reply letter).

Reviewer #2 (Remarks to the Author):

In this version the authors have addressed the concerns clearly and appropriately.

Reviewer #3 (Remarks to the Author):

The revised MS has answered all the questions raised by the review one by one. A lot of experiments were supplemented, and sufficient evidence were provided to confirm the previous doubts. However, there is one point that the author needs to further explain:

It is likely that Rv2780 directly inhibits NF-kB activation. The author supplemented the experiment for examining inflammation cytokines expression (IL-1 β , IL-6, IL-12 and Tnf- α) in lung tissues infected with H37Rv and Δ Rv2780. Results showed that Rv2780 promotes cytokines expression in macrophages or lung tissues upon M. tuberculosis infection (revised Supplementary Fig. 2I-P), This is odd, because if Rv2780 directly inhibits NF-kB activation, then these cytokines should also be inhibited rather than elevated, how to explain this result?

REVIEWER COMMENTS

Reviewer #1 (Remarks to the Author):

The authors invested substantial efforts to improve the manuscript.

We thank the reviewer for appreciation of our efforts and the positive affirmation of our revision.

1. Statistics and methodology related to normalization of alanine concentrations in macrophages should be provided in the manuscript (materials and methods, legend Fig. 2 G,H).

As per the reviewer's suggestion, we have provided statistics and methodology related to normalization of alanine concentrations in macrophages in the revised manuscript (materials and methods (**Line 776 - Line 781**), legend Fig. 2 G, H (**Line 1242 and Line 1247**)). In our study, we detected alanine concentration of macrophages by Alanine Assay Kit (MAK001, Sigma-Aldrich), and alanine level was normalized to GAPDH, an indicator of the number of cells. GAPDH quantification were calculated by gray scale analysis of immunoblotting of GAPDH from macrophages. Then the normalized alanine level of each sample shown in the figure was calculated as followings: Normalized alanine level = Alanine concentration / Gray value of GAPDH.

2. The Prss1 KO phenotype is likely strongly NF-κB-related and not only AMP-related. Accordingly, the observed very strong phenotype in TB (and other infections) (Fig. 4). These observations should be at least acknowledged and accordingly discussed.

We highly appreciate the reviewer's insightful comments. Upon infection of pathogens, NF-κB signaling is activated to induce the expression of not only antimicrobial peptide genes, but also pro-inflammatory genes including cytokines and chemokines¹. Our *in vitro* study showed that *Prss1*^{+/-} mice peritoneal macrophages infected with *M. tuberculosis* H37Rv had much lower intracellular CFU than those WT counterparts (**Figure 4N-O**). Given that antimicrobial peptides (AMPs) directly target intracellular bacteria, *Prss1* deficiency may mediate bacterial clearance *in vitro* mainly through regulating the expression of AMPs. As per the reviewer's suggestion, we have discussed this in the revised manuscript (**Line 485 - Line 489**).

Our *in vivo* study showed that significantly decreased lung bacterial burden and tissues damages were observed in *Lyz2*^{cre}*Prss1*^{flox/flox} mice infected with *M. tuberculosis* (**Figure 4K-M**), suggesting that *Prss1* may negatively regulate anti-TB immunity through downregulating the activation of NF-κB signal (**Figure 4I**) and not only AMP-related. Because higher NF-κB activation may not only induce AMPs expression, but also promote cytokines and chemokines expression in *Prss1* deficient macrophages, which may subsequently activate other immune cells (including neutrophils and T cells) to maintain the *in vivo* anti-TB immunity. Therefore, we could not exclude the involvement of other NF-κB regulated genes, which needs further exploration. As per

the reviewer's suggestion, we have discussed this in the revised manuscript (**Line 489 - Line 498**).

Besides, NF- κ B plays a central role in host response to different infection of pathogens^{2,3}, including gram-negative bacteria *E. coli* or gram-positive bacteria *S. aureus* or *M. tuberculosis*. Thus, the observation that the inhibitory effects of Prss1 on NF- κ B activation were also observed during other bacterial infections (**Figure 4I; Supplementary Fig. 7G-H**). As per the reviewer's suggestion, we have discussed this in the revised manuscript (**Line 501 - Line 504**).

3. The titles of the figures should be carefully checked for accuracy. Fig. 1 does not show that Rv2780 inhibits AMPs, rather that the mutant is attenuated in macrophages and mice. The title for Fig. 4 is a statement that the authors themselves indicated as non-accurate "L-alanine interacts with PRSS1 to induce Defb4 expression" (similar to Fig. S7 L-alanine interacts with PRSS1 to induce AMPs) (see authors' reply letter).

As per the reviewer's suggestion, we have checked the title of each figure very carefully. The title of Fig. 1 has been changed to "Rv2780 is a virulence factor" (**Line 1209**). As for the title of **Fig. 4** we use a more accurate statement that "L-alanine interacts with Prss1 to induce NF- κ B-mediated *Defb4* expression" (**Line 1288 - Line 1289**). The title of **Fig. S7** we use a more accurate statement that "L-alanine interacts with Prss1 to induce NF- κ B-mediated AMPs expression" (**Supplementary information - Line 115**).

Reviewer #2 (Remarks to the Author):

1. In this version the authors have addressed the concerns clearly and appropriately.

We thank the reviewer for the positive affirmation of our revision.

Reviewer #3 (Remarks to the Author):

1. The revised MS has answered all the questions raised by the review one by one. A lot of experiments were supplemented, and sufficient evidence were provided to confirm the previous doubts. However, there is one point that the author needs to further explain:

We thank the reviewer for the appreciation of our efforts and positive affirmation of our revision.

2. It is likely that Rv2780 directly inhibits NF- κ B activation. The author supplemented the experiment for examining inflammation cytokines expression (IL-1 β , IL-6, IL-12 and Tnf- α) in lung tissues infected with H37Rv and Δ Rv2780. Results showed that Rv2780 promotes cytokines expression in macrophages or lung tissues upon *M. tuberculosis* infection (revised Supplementary Fig. 2I-P), This is odd, because if Rv2780 directly inhibits NF- κ B activation, then these

cytokines should also be inhibited rather than elevated, how to explain this result?

We appreciate the reviewer's insightful question. Although the expression of pro-inflammatory cytokines is mainly regulated by NF- κ B activation, there are many NF- κ B-independent mechanisms regulating cytokines expression such as epigenetic regulation of microRNA^{4,5} or histone modification⁶. Besides, transcription factor Nrf2 suppresses inflammation through redox control without affecting NF- κ B activation⁷. Thus Rv2780 may elevate inflammation cytokines expression through other NF- κ B-independent pathways, which needs further investigation. The discussion part of the manuscript has been revised as stated above (**Line 511 - Line 519**).

References

1. Liu, T., Zhang, L., Joo, D. & Sun, S. C. NF- κ B signaling in inflammation. *Signal Transduct Target Ther.* 2, 17023- (2017).<https://doi.org/10.1038/sigtrans.2017.23>
2. Xu, Y. R. & Lei, C. Q. TAK1-TABs Complex: A Central Signalosome in Inflammatory Responses. *Front. Immunol.* 11, 608976 (2020).<https://doi.org/10.3389/fimmu.2020.608976>
3. Rahman, M. M. & McFadden, G. Modulation of NF- κ B signalling by microbial pathogens. *Nat Rev Microbiol.* 9, 291-306 (2011).<https://doi.org/10.1038/nrmicro2539>
4. Zuo, X., Wang, L., Bao, Y. & Sun, J. The ESX-1 Virulence Factors Downregulate miR-147-3p in *Mycobacterium marinum*-Infected Macrophages. *Infect. Immun.* 88 (2020).<https://doi.org/10.1128/iai.00088-20>
5. Singh, Y. et al. *Mycobacterium tuberculosis* controls microRNA-99b (miR-99b) expression in infected murine dendritic cells to modulate host immunity. *J. Biol. Chem.* 288, 5056-5061 (2013).<https://doi.org/10.1074/jbc.C112.439778>
6. Yang, H. et al. Interception of host fatty acid metabolism by mycobacteria under hypoxia to suppress anti-TB immunity. *Cell Discov.* 7, 90 (2021).<https://doi.org/10.1038/s41421-021-00301-1>
7. Kobayashi, E. H. et al. Nrf2 suppresses macrophage inflammatory response by blocking proinflammatory cytokine transcription. *Nat. Commun.* 7, 11624 (2016).<https://doi.org/10.1038/ncomms11624>